# An ultrafast diamond nonlinear photonic sensor

Daisuke Sato[1,4], Junjie Guo [1,4], Takuto Ichikawa [1], Dwi Prananto [2], Toshu An [2], Paul Fons [3], Shoji Yoshida[1], Hidemi Shigekawa [1] & Muneaki Hase [1] ✉

The integration of light and materials technology is key to the creation of innovative sensing technologies. Sensing of electric and magnetic fields, and temperature with high spatio-temporal resolution is a critical task for the development of the next-generation of nanometer-scale quantum devices. Color centers in diamonds are attractive for potential applications owing to their characteristic quantum states, although they require metallic contacts for the introduction of external microwaves. Here, we build an ultrafast diamond nonlinear photonic sensor to assess the surface electric field; an electro-optic sensor based on nitrogen-vacancy centers in a diamond nanotip breaks the spatial-limit of conventional pump-probe techniques. The 10-fs near-infrared optical pulse modulates the surface electric field of a 2D transition metal dichalcogenide and we monitor the dynamics of the local electric field at nanometer-femtosecond spatio-temporal resolutions. Our nanoscopic technique will provide new horizons to the sensing of advanced nano materials.

The electro-optic (EO) effect is a nonlinear optical phenomenon in which the optical properties of a material are affected and modified by an electric field[1]. Transparent materials such as ferroelectrics are mainly used as nonlinear optical materials, and since the invention of the laser, they have been widely applied to optical communications[2], optical devices, and optical circuits such as optical switches[3], modulators[4], and sensors[5]. As sensors, the first-order EO effect, the Pockels effect, has been used to measure the electric fields of electromagnetic waves in free space[5] and in electric circuits that are in contact with a nonlinear optical material with high time-resolution on the order of sub-picoseconds[6]. However, with existing electro-optic sensors, the spatial resolution is determined by the diffraction limit of the light, and nanometer resolution has been difficult to achieve.

Within modern quantum technologies, color centers in diamond have become an indispensable element of sensing systems for the detection of localized magnetic and electric fields and temperature[7–14]. In particular, the nitrogen-vacancy (NV) center has enabled the development of the ultimate quantum sensor based on optically detected magnetic resonance (ODMR)[7–9] and has attracted considerable attention owing to its potential applications in quantum information processing[15,16], quantum communication[17,18], and quantum networks[19]. Although nanometer resolution quantum sensing using diamond nanoprobes has been established, the time resolution of the conventional technique based on ODMR remains in the nanosecond regime[8,13,14,20,21], making high time-resolution detection a challenge.

Since pure defect-free diamonds are colorless and their crystal structure has spatial inversion symmetry, their second-order nonlinear susceptibility $\chi^{(2)}$ is strictly zero, and second-order nonlinear optical effects such as the Pockels effect do not occur[22–25]. To overcome this disadvantage, we have used diamond crystals with NV centers, where the second-order nonlinear susceptibility is non-zero ($\chi^{(2)} \neq 0$) due to the breaking of spatial inversion symmetry by NV defects[26], and hence, the Pockels effect is expected to occur[27]. Since the electro-optic effect does not require real charge carrier excitation, i.e., non-resonant transitions, our 1.56 eV photon does not directly interact with the NV spin states (Supplementary Note 1). On the other hand, structured illumination microscopy (SIM) is a commercially available super-

[1]Department of Applied Physics, Institute of Pure and Applied Sciences, University of Tsukuba, Tsukuba, Ibaraki, Japan. [2]School of Materials Science, Japan Advanced Institute of Science and Technology, Nomi, Ishikawa, Japan. [3]Department of Electronics and Electrical Engineering, Faculty of Science and Technology, Keio University, Yokohama, Kanagawa, Japan. [4]These authors contributed equally: Daisuke Sato, Junjie Guo. ✉e-mail: mhase@bk.tsukuba.ac.jp

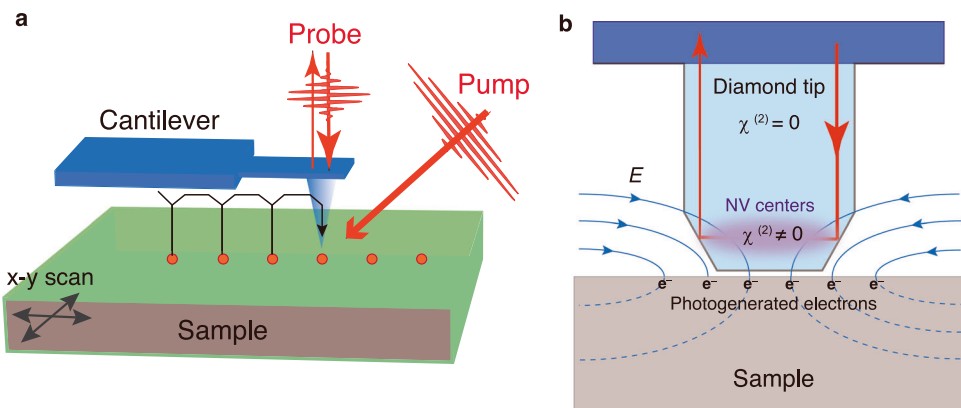

**Fig. 1 | Concept of the electro-optic nanoscopy. a** Schematic of the ultrafast pump-probe sensing measurement with a diamond NV tip with the "pin-point mode", that is vertically approaching and retracting the AFM probe at each designated points on the sample. The sample is scanned in the x-y direction using a piezo-scanner. **b** Schematic of electro-optic sampling using a diamond NV tip. The Pockels effect occurs where the spatial inversion symmetry is broken by the NV centers ($\chi^{(2)} \neq 0$), and the refractive index change ($\Delta n \propto \chi^{(2)}_{ij}\Delta E(t)$) of this part is optically probed. The electric field, which is detected by the NV probe, is modulated by the photogenerated electrons at the sample surface.

resolution microscopy, with a sub-micrometer resolution. Although time-resolved SIM has also been reported[28], the achievable time resolution is seconds or greater. There is thus a still strong motivation to develop a new technique with ultrafast time resolution.

Here, by combining an optical pump-probe technique with scanning probe microscopy (SPM) technology[29], we developed an ultra-precise spatio-temporal sensing technique with ≤500 nm and ≤100 fs spatio-temporal resolutions. Moreover, by introducing the NV defects at a shallow (<40 nm) depth from the diamond surface, we show in the following a highly surface-sensitive SPM probe based on the Pockels effect (Fig. 1). Our idea is to excite the sample with sufficiently short light pulses and to sense electric fields in the sample through the Pockels effect of the surface-sensitive diamond nanoprobes as a function of the time delay $t$ (Fig. 1b).

## Results

### Diamond nonlinear photonic sensor

The diamond NV probe was fabricated using laser cutting and a Ga$^+$ ion focused ion beam (FIB) milling technique that is described in ref. 30 (see also "Methods"). A (100)-oriented CVD-grown electronic-grade bulk diamond single crystal with initial nitrogen impurities of <5 ppb was used. Photoluminescence (PL) measurements indicated that the electronic state of the NV diamond was a mixture of negatively charged states (NV$^-$) and neutrally charged states (NV$^0$); however, the charge state of NV$^-$ would play a central role in the strong enhancement of the EO effect as presented in ref. 27 (see also Supplementary Note 2). The spatial resolution of the diamond NV probe is better than ≈ 660 nm and potentially ≤500 nm because of the enhancement of the EO sensitivity for the apex of the NV tip, based on scanning ion microscopy measurements and PL images (Fig. 2a–c) (Supplementary Note 3).

The fabricated diamond NV probe was successfully attached to the tip of a self-sensing cantilever (Fig. 2d–f), where atomic forces can be detected by means of a piezoelectric sensor[31]; the EO effect of the diamond NV probe was evaluated under this setup. The optical system around the microscope was constructed and optimized using a concave mirror to inject the excitation pulse light from a 45-degree oblique direction to the probe, while a reflective long-focus objective lens was used to inject only the probe light from the back side of the tip (Supplementary Notes 4 and 5)[32]. A reflective (Schwarzschild) type objective lens was selected to minimize the dispersion of the Ti: sapphire pulse laser[33], whose pulse length was ≤10 fs and whose wavelength covered the region from 660 to 940 nm (1.88 to 1.32 eV). Using an AFM system with a self-sensing cantilever, we focused the

probe beam on the back of the cantilever, specifically on the diamond NV tip.

### Electro-optic sampling on the surface of a semiconductor

We first evaluated the sensitivity of the nonlinear optical response of the diamond NV probe using a prototypical semiconductor bulk wafer, n-type semiconducting GaAs, as a test sample whose band-gap energy is ≈1.5 eV at room temperature[34,35]. Because of the high density of surface states, the Fermi level is pinned mid-gap, leading to band bending (Fig. 3a)[35]. In an n-type semiconductor, the bands bend upward, resulting in a depletion region near the surface ($\lambda_d \sim 100$ nm depth) and a static electrical field $E$ perpendicular to the surface[34]. Note that the surface electric field is determined by the potential of the surface states, enabling EO measurements with the NV tip as demonstrated below. Upon the photoexcitation of an n-type semiconductor, electron-hole pairs are generated near the surface and screen the surface electrical field, resulting in a change of $E$, $\Delta E$. Here, the EO response can be measured by a change in anisotropic reflectivity, which is proportional to the surface electric field[34],

$$\frac{\Delta R_{eo}(t)}{R_0} = \frac{4r_{ij}n_0^3}{n_0^2 - 1}\Delta E(t), \tag{1}$$

where $\Delta R_{eo}(t)$ is the anisotropic reflectivity change and $R_0$ is the reflectivity without photoexcitation, $r_{ij}$ is the electro-optic (Pockels) coefficient and $n_0$ is the refractive index, both of which are material dependent (see Supplementary Note 4 in more details). In the case of GaAs[36,37], for instance, $r_{41} = -1.6$ pm V$^{-1}$ and $n_0 = 3.7$. A positive EO response with a picosecond relaxation time observed under macroscopic conditions (Fig. 3b) indicates upward band bending near the surface, a characteristic phenomenon for n-type GaAs[34], and indicates that the surface electric field can be accurately measured. The ≈1.1 ps relaxation dynamics is expected to be governed by the change in the surface electric field, i.e., the field is initially screened by the photogenerated carriers and later recovers after carriers relax via scattering with phonons or diffusion out of the depletion region[34]. Using Eq. (1), we obtain $\Delta E \approx -3.1 \times 10^6$ V m$^{-1}$ from the maximum experimental EO response value ($\Delta R_{eo}/R_0 = 2.2 \times 10^{-4}$) under macroscopic conditions, which is in good agreement with that obtained for n-GaAs by a similar technique[34,38]. Moreover, for the case of the EO response under the NV tip as shown in Fig. 3c, although the EO signal amplitude decreases to ≈1/42 of the macroscopic case (Fig. 3b), we still observe a positive EO signal with ≈0.5 ps relaxation time, indicating that sensing measurement is possible through the diamond NV tip (see

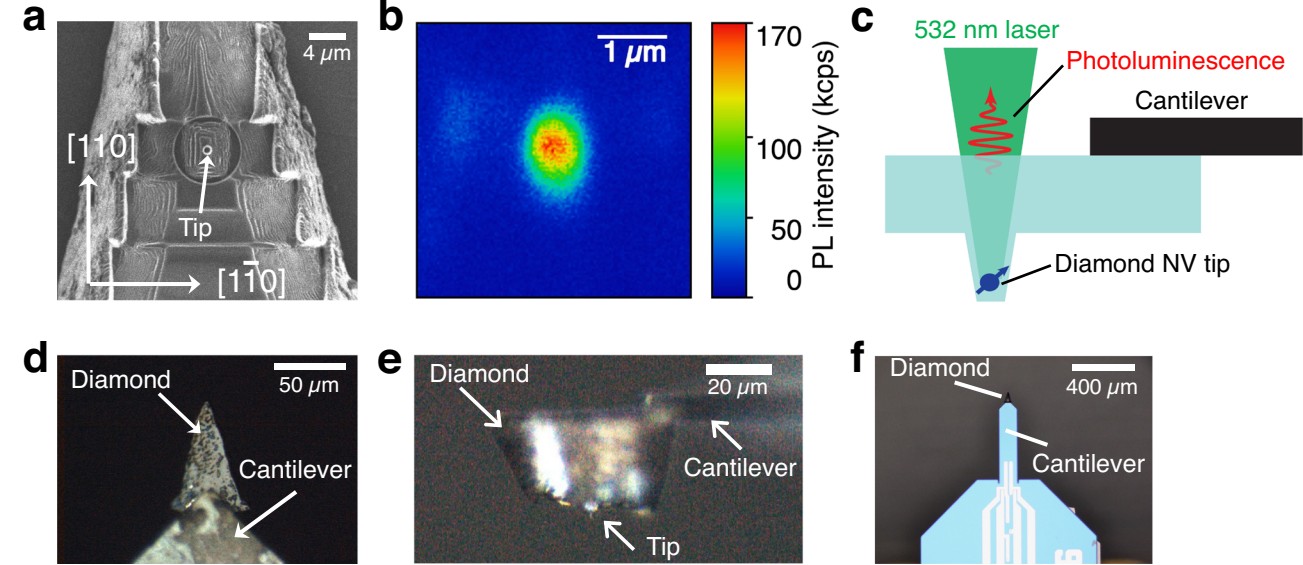

**Fig. 2 | The fabrication of the diamond NV probe. a** Scanning ion microscopy image of the FIB-fabricated diamond NV tip. **b** Photoluminescence image of the diamond NV tip. **c** Schematic of photoluminescence measurement to confirm the NV centers in the diamond tip. The photoluminescence was collected by an avalanche photodiode through a confocal microscope after the irradiation of a 532 nm cw laser at 135 μW. **d** Top view optical image of the diamond NV probe attached to the cantilever. **e** Side view of the diamond NV probe attached to the cantilever. **f** Optical image of the self-sensing cantilever with a diamond NV probe.

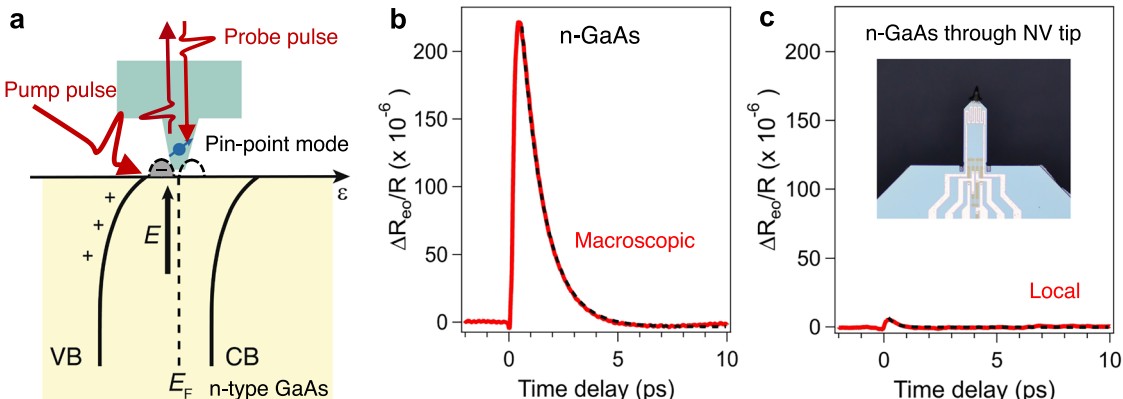

**Fig. 3 | Sensing of electric field on n-GaAs with the diamond NV probe.** **a** Schematic of surface band bending and layout of the pin-point mode of the diamond NV tip. The surface states are presented by bell-shaped dashed lines with the Fermi energy ($E_F$), where the lower band is occupied by electrons (−). VB and CB denote valence and conduction bands, respectively. **b** Macroscopic time-resolved EO signal from an n-GaAs wafer without the diamond NV probe observed at room temperature. **c** Local time-resolved EO signal from n-GaAs with the diamond NV probe. The inset shows the optical image of the self-sensing cantilever with a diamond NV probe. The black dashed curves in **b** and **c** represent exponential fits to extract the relaxation time constant explained in the main text.

Supplementary Note 6 for the value of the Pockels coefficient of NV diamond).

If the probing area is limited by the apex of the NV probe and this causes a decrease of the EO signal by ≈1/42 in Fig. 3, we can estimate the diameter of the NV probe to be ≈800 nm, which is consistent with the observation shown in Fig. 2 (see also Supplementary Note 3). The nearly halving of the relaxation time constant for the case of the NV tip suggests carrier scattering and diffusion in the surface region is stronger due to surface defects and/or 2D conduction[39,40]. Based on these observations, we performed electric field sensing for two-dimensional layered materials to further demonstrate the potential applicability of the diamond NV probe to advanced materials investigations as described below.

**Electro-optic sampling on a transition metal dichalcogenide**
To demonstrate that it is possible to measure the surface electric field with nanometer-resolution on a monolayer material in addition to bulk materials, micrometer-sized monolayer (1 ML) to multiple layers (bulk)

of the transition metal dichalcogenide (TMDC) WSe₂ on a Si substrate covered with a 100-nm SiO₂ layer were prepared from a single crystal by using the Au assisted transfer method (see "Methods")[41]. The regions containing both 1 ML and bulk of WSe₂ were confirmed by micro-Raman measurements by the observation of characteristic phonon modes (Supplementary Note 7)[42–44]. The A-exciton direct band-gap (K-point) energy of the 1 ML WSe₂ on SiO₂/Si is ≈1.67 eV at room temperature, while that of the bulk is ≈1.63 eV (ref. 45). Since these energies are both covered by the broadband 10-fs laser used, electron-hole pairs are photogenerated to form A-excitons and subsequently dissociate due to the Mott transition within ~100 fs in both 1 ML and bulk WSe₂ samples[46]. Note that the pump fluence of 215 μJ cm⁻² generates a carrier density of ≈1.35 × 10¹³ cm⁻² per layer, being above the threshold of the expected Mott transition (≈3 × 10¹² cm⁻²)[47]. Note also that bulk WSe₂ has an indirect gap of ≈1.2 eV at the Λ point[48], leading to a significantly different electron thermalization process.

Before the EO measurements, the morphology of the WSe₂ sample was characterized under an optical microscope as well as AFM

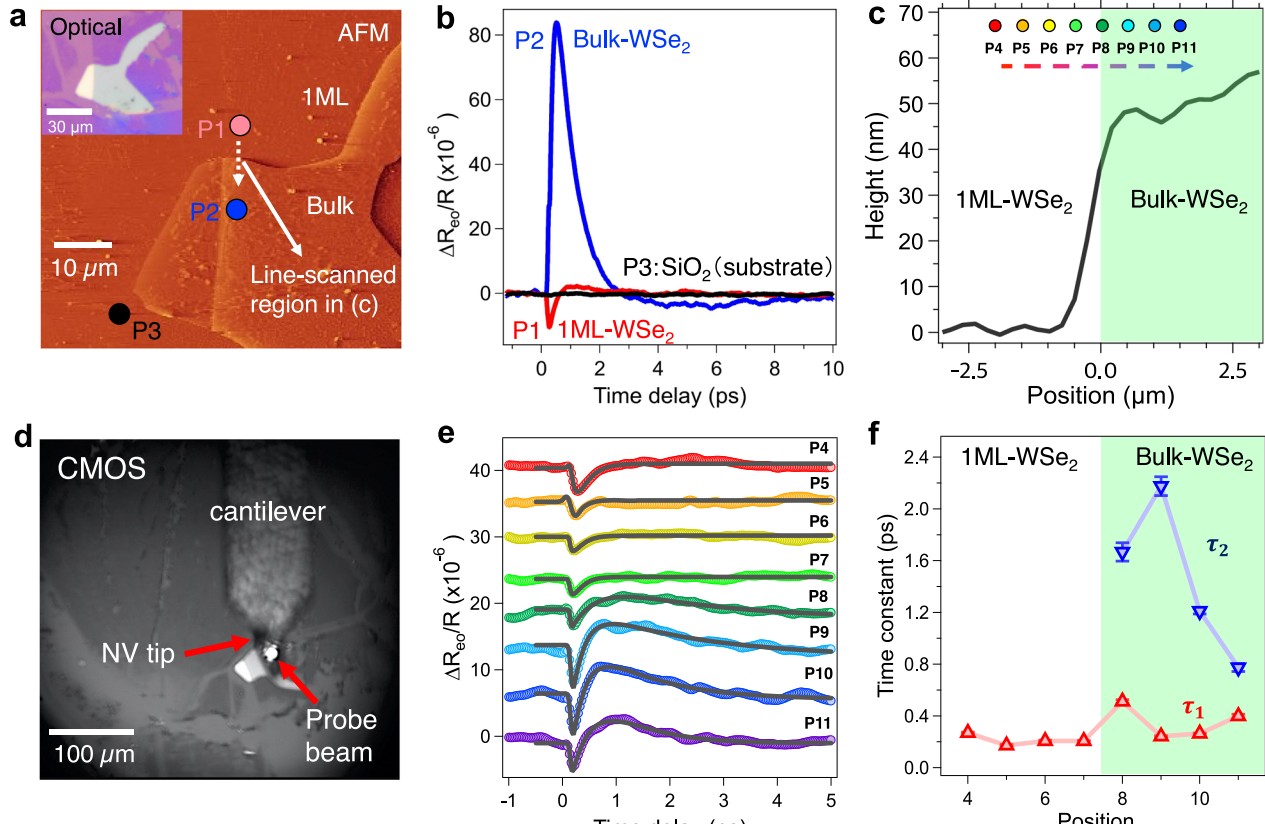

**Fig. 4 | Spatio-temporal measurements of EO signal on WSe₂. a** Topographic image of the 60 μm × 60 μm region with the diamond NV probe. P1 - P3 denote the macroscopic measurement positions, P1 refers to the 1 ML WSe₂, P2 to the bulk WSe₂, and P3 to the SiO₂ substrate. The dashed arrow represents the line-scanned region in (**c**) and (**e**). **b** Time-resolved EO signals obtained under the macroscopic conditions (without diamond NV probe) from three different positions on WSe₂/SiO₂ at room temperature. **c** AFM line scan at 200 nm step using a silicon tip. P4 - P11 represent the measurement positions, from the monolayer to the multiple layers (bulk) of WSe₂. The fluctuation of height is possibly due to surface wrinkles of the sample and/or vibrations from the clean booth system. **d** CMOS image of the diamond NV probe and the sample. **e** Local time-resolved EO signal obtained with the diamond NV probe. The black solid lines in **e** represent the fits using a single (1 ML) or double exponential (bulk) decay function. **f** Time constants obtained by the fit to the time-resolved EO signals at different positions. Error bars represent the standard deviations. The rectangle green shading in **c** and **f** represents the bulk region.

(Fig. 4a), and we focused on the interface between the 1 ML region and bulk WSe₂. First, using macroscopic EO spectroscopy (without the diamond NV probe), we obtained a negative EO signal in the monolayer region and a positive EO signal from bulk WSe₂ (Fig. 4b). In the bulk region, the positive EO response showed an exponential relaxation time of $\tau \approx 0.8 \pm 0.1$ ps, whereas in the 1 ML region, the negative EO response exhibited a relaxation time of $\tau \approx 0.3 \pm 0.1$ ps. The time constants are consistent with those of intraband and intervalley scattering obtained for WSe₂ by time-resolved angle-resolved photoemission spectroscopy measurements[49]. The faster relaxation time in the 1 ML region suggests additional relaxation pathways of the photogenerated electrons via stronger coupling between the 1 ML WSe₂ and the SiO₂ substrate or scattering by surface defects[50]. Thus, we successfully obtained EO signals that reflect the characteristics of the carriers (see "Methods"). With macroscopic measurements (Fig. 4b), we observed the bulk region at an optical penetration depth $\lambda_p \sim 50$ nm (see "Methods"), while the observation was dominated by the surface region for the case of the NV tip with the AFM. The surface of our WSe₂ sample was p-type, where holes dominate the electronic properties, due to surface oxidation[51,52], while the bulk region is n-type or intrinsic, where electrons dominate the electronic properties. The difference in the depth information is expected to affect the EO signal as described below.

From a line scan of topography data using a Si cantilever, the height of the bulk WSe₂ was found to be ~50 nm (Fig. 4c), matching the expected penetration depth $\lambda_p$. Furthermore, we performed

nanometer-scale local EO measurements by contacting the sample with the diamond NV tip and obtained EO signals reflecting the local electric field of the sample (Fig. 4d, e). For 1 ML WSe₂, we observed negative EO signals, demonstrating the p-type character of the sample nature due to surface oxidation. In contrast, for the bulk WSe₂ sample, we observed negative EO signals, i.e., a p-type signal, at $t \approx 0$ ps but interestingly the EO signal changed to a positive signal at $t \approx 1$ ps (Fig. 4e). It is also interesting to investigate the relaxation time constant across the boundary (Fig. 4f). In the 1 ML region, the negative EO response shows a single exponential relaxation time of $\tau \approx 0.2 \pm 0.1$ ps (P4), while in the bulk region it shows a double exponential relaxation: the initial negative signal exhibits $\tau_1 \approx 0.3 \pm 0.1$ ps while the second positive signal decay had value of $\tau_2 \approx 2.2 \pm 0.1$ ps (P9), a value much longer than that of the negative EO signal as well as that observed without the NV probe. Note that the time resolution under the NV tip can be estimated to ≈100 fs from the full width at the half maximum (FWHM) of the shortest EO response observed at the position of P9.

To further analyze the observed dynamics on the bulk WSe₂, we examine the time evolution of the carrier density $N$ based on the Boltzmann transport equation[53]:

$$\frac{\partial N}{\partial t} = -N/\tau_{ep} - BN^2 - \gamma N^3 + D\nabla^2 N, \qquad (2)$$

where $\tau_{ep}$ is the intraband and/or intervalley scattering time constant, $B$ denotes the radiative recombination coefficient, $\gamma$ is the Auger recombination coefficient, and $D$ is the ambipolar diffusion coefficient. Under the assumptions that the photoexcited carrier density exceeds the threshold for the Mott transition and excitonic Auger processes can be neglected[54], the right-hand side of Eq. (2) just after the photoexcitation (before radiative recombination occurs on ps ~ ns time scales[46]) can be simplified to $-N/\tau_{ep} + D\nabla^2 N$. Here using $D = \left\langle \frac{v^2 \tau_m}{2} \right\rangle$ with $\tau_m$ the momentum relaxation time and the Einstein's relation $\frac{D}{\mu} = \frac{k_B T}{e}$, with $k_B$ the Boltzmann constant, $T$ the temperature and $e$ the electron charge, we obtain $D = 0.56\,\mathrm{cm^2 s^{-1}}$ assuming the electron mobility is $\mu = 30\,\mathrm{cm^2 V^{-1} s^{-1}}$ at room temperature[55]. We then find the electron velocity $\langle v \rangle = 3.35 \times 10^4\,\mathrm{m\,s^{-1}}$ for $\tau_m \approx 100\,\mathrm{fs}$ (ref. 33). This means that the photogenerated electrons disappear from the surface region ($\leq 1\,\mathrm{nm}$) within $\approx 0.3\,\mathrm{ps}$, a period being matched with $\tau_1$ ($0.3 \pm 0.1\,\mathrm{ps}$).

Note that the second time constant ($\tau_2 \approx 2.2\,\mathrm{ps}$) is nearly consistent with the average time for phonon emissions by carrier cooling in the $\Lambda$ valley, i.e., the intraband scattering time $\tau_{ep} \approx \tau_2$, if we consider the emission of the optical $A_{1g}$ mode whose frequency is 7.5 THz; $0.133\,\mathrm{ps} \times 16$ emissions (Supplementary Note 8)[56]. The intraband scattering in the $\Lambda$ valley will be, however, hindered by direct intervalley scattering from the K to $\Lambda$ valleys ($\leq 0.5\,\mathrm{ps}$)[49], which will contribute to the initial decay of the EO signal through population by nonequilibrium electrons at the $\Lambda$ valley ($\tau_1 \approx 0.3\,\mathrm{ps}$). This path is followed by trapping by the surface defect states[40] ($\tau_2 \approx 2.2\,\mathrm{ps}$). Thus, we are able to explain our observation of the dynamics of bulk WSe$_2$ as the combination of intervalley scattering and trapping by surface defect states[40] along the K-$\Lambda$ valleys (Supplementary Note 8). Since the surface electric field is sensitive to the density of defects, possible targets for the measurement of local electric fields will potentially include power device materials such as SiC, as well as topological insulators and other TMDCs with mono-, bi-, and tri-layers.

In our experiment, contrary to an ODMR experiment, the laser energy (1.56 eV) was not resonant with transitions in the NV states ($A_2 \rightarrow E_1$ and $A_2 \rightarrow E_2$) (Supplementary Note 1), and therefore, is not related to the conventional dc Stark effect[12]. However, our nonlinear optical method may be related to the "optical" Stark effect, known as the inverse Faraday effect (IFE) or inverse Cotton-Mouton effect (ICME), in which light-induced dc magnetization arises because the optical field shifts the different magnetic states of the ground manifold differently and mixes into these ground states different amounts of excited state[1]. We have, in fact, observed both IFE and ICME in NV diamond[57], implying the "optical" Stark effect may also be realized using a diamond NV probe in the near future.

Note that the general sensitivity based on measurement time (typically 1 s at a 10 Hz scanning frequency) and the noise level of $\Delta R_{eo}/R_0 \approx 2 \times 10^{-7}$ corresponding to $|\Delta E| \approx 9.6 \times 10^{-3}\,\mathrm{V\,\mu m^{-1}}$ or $96\,\mathrm{V\,cm^{-1}}$ for the EO signal in n-GaAs can be used to discuss the resolution. Using the above parameters, we could estimate the sensitivity to be $\sim 1 \times 10^{-2}\,\mathrm{V\,\mu m^{-1}\,Hz^{-1/2}}$ or $\sim 100\,\mathrm{V\,cm^{-1}\,Hz^{-1/2}}$, which is comparable to that of the conventional NV-based electrometer by Dolde et al.[11], but three-orders worse than recent work by Michl et al.[58]. Note also that there exist commercially available NV diamond tips from QZabre or Qnami[59], with which we may improve the spatial resolution down to $\sim 10\,\mathrm{nm}$ if the sensitivity is increased down to the scale of several or an even single NV center in future experiments. In addition, the development of a diamond nonlinear photonic sensor system to be used in vacuum, and operating in non-contact mode, i.e., an oscillating tip that maintains a constant distance between the tip and sample, might also be useful for advancing EO-based electrometer technology regarding the measurement of the vertical distribution of the electric field.

To summarize, we proposed and realized spatio-temporal measurements of the surface electric field on a two-dimensional transition metal dichalcogenide by taking advantage of an electro-optic sensor based on NV centers in a diamond nanotip combined with a 10-fs pulsed laser. High sensitivity of the NV nanotip for electro-optic sampling was demonstrated on a doped semiconductor wafer. Compared with conventional macroscopic measurements, the local sensing method provides a higher sensitivity for the surface probe, resulting in an excellent information limit with respect to time ($\leq 100\,\mathrm{fs}$) and spatial ($\leq 500\,\mathrm{nm}$) resolutions and observation of the carrier screening dynamics at the surface of bulk transition metal dichalcogenide. By further developing the sensitivity of the NV probe to the single NV level, it can be surmised that sensing measurements at a spatial resolution down to $\approx 10\,\mathrm{nm}$ spatial resolution can be achieved. Since the NV probe is sensitive to both spins[59] and temperature[60], our approach will provide additional degrees of freedom to detect magnetic and thermal fields, in addition to the sensing of the electric field. Given the capabilities and considering the improved spatio-temporal limit, our proposed technique should find wide applications in materials science and nanotechnology.

## Methods
### Fabrication of nano probe
The NV centers were created at a precisely controlled density by means of $^{14}N^+$ ion implantation (incident energy 30 keV), followed by annealing at 900 °C for 1 h in Ar atmosphere. The implanted depth deduced from the Monte Carlo simulation (Stopping and Range of Ions in Matter: SRIM[61]) was about $\sim 40\,\mathrm{nm}$ and the profile was close to a Gaussian function with FWHM of $\sim 25\,\mathrm{nm}$ (Supplementary Note 9). Photoluminescence measurements indicated that the electronic state of the NV diamond was a mixture of negatively charged states (NV$^-$) and neutrally charged states (NV$^0$), indicated by the observation of the zero-phonon line (ZPL) at 638 nm and 575 nm for NV$^-$ and NV$^0$, respectively, and broad phonon sideband at $\sim 680\,\mathrm{nm}$ and $\sim 620\,\mathrm{nm}$ for NV$^-$ and NV$^0$, respectively[26,27]. By introducing NV centers into high-purity diamond single-crystals, we have succeeded in obtaining an EO effect enhancement of $\approx 13$ times for $1 \times 10^{12}\,\mathrm{cm^{-2}}$ $^{14}N^+$ implantation dose with respect to that before ion irradiation (pure diamond) (ref. 27). This optimal density of the NV center was confirmed by ODMR measurements. Thus, the EO effect can be controlled by the appropriate implantation dose of $^{14}N^+$ ions, and we applied the optimal density for the fabrication of the diamond NV probe. Note that the number of the NV centers included in the diamond NV probe can be estimated to be $\sim 710$, assuming a tip volume of $3.14 \times 10^{-2}\,\mathrm{\mu m^3}$ and a density of the NV centers of $3.6 \times 10^{16}\,\mathrm{cm^{-3}}$ (ref. 27). The polarization of the pump and probe beams were set in the [100] direction and the [1$\bar{1}$0] direction, respectively, on the diamond tip. Note also that other quantum defects in diamond, such as SiV (or GeV, etc.) centers[62], do not break inversion symmetry (Supplementary Note 10). Thus, the NV center is one of the color centers in diamonds that enables EO sensing capability, although other color centers, such as boron-vacancy (BV), oxygen-vacancy (OV) centers, may also break inversion symmetry[63].

### Sample preparation
The monolayer (1 ML) WSe$_2$ sample was prepared by Au-assisted transfer methods using polymethyl methacrylate (PMMA)[41]. In the process of cleaning WSe$_2$, it is first cleaned with water, and then PMMA is dissolved in acetone. The doping effect of the organic solvents used has been experimentally confirmed, and acetone has an electron-doping effect on WSe$_2$. In addition, for the case of WSe$_2$, the material is a little more vulnerable to environment-induced degradation than MoS$_2$, so the surface oxidation effects are expected, which transform the surface electronic carriers to p-type[51,52]. The regions of the 1 ML and bulk of WSe$_2$ were confirmed by micro-Raman measurements by the observation of the appropriate optical modes.

## Ultrafast nanoscopy

The electric field response was measured by an electro-optic (EO) sampling method based on a reflective pump-probe scheme. The light source was an ultrashort pulse femtosecond oscillator (Element 2, Spectra-Physics), which generates ≤10 fs pulses with a central wavelength of 795 nm (1.56 eV) at a repetition rate of 75 MHz. The time delay between the pump and probe light was modulated at a frequency of 10 Hz and an amplitude of 15 ps by an oscillating retroreflector (shaker) placed in the pump light path. The pump light was focused onto the sample by a 90-degree off-axis parabolic mirror with a focal length of 50.8 mm, while the probe was focused by a reflective (Schwarzschild) type objective lens from Thorlabs, whose working distance is 23.8 mm. Assuming the incident beam diameter is 4 mm, the pump light with an average power of 90 mW and 9 mW probe light correspond to the fluences of 215 μJ cm$^{-2}$ and 22 μJ cm$^{-2}$, respectively. The optical penetration depth at 795 nm was estimated to be ~50 nm for $WSe_2$ based upon the reported absorption coefficient of ≈$2 \times 10^5$ cm$^{-1}$ (ref. [64]). The photodetector used was a 1 GHz high-speed Si-PIN photodetector (S5973-01) from Hamamatsu Photonics. The random noise of the signal on each scan was reduced by integrating the signal with a digital oscilloscope with the number of accumulations used being 5000. The self-sensing AFM system was built by Anfatec Instruments AG using a piezo-sensitive cantilever from SCL-Sensor Tech. The cantilever has a length of 450 μm, a width of 100 μm, and a thickness of 10 μm. We use the atomic force microscopy (AFM) "pin-point mode", that is, vertically approaching and retracting the AFM probe at designated points on the sample as shown in Fig. 1a (see Supplementary Note 11 for the force-distance curve). Thus, the distance of the diamond probe from the sample surface was quasi-zero at each of the designated points, which was determined by the Lennard-Jones potential (Typically 0.1–0.3 nm)[65].

## Data availability

All data that support the findings in this paper are available within the article and its Supporting Information or are available from the corresponding authors upon request. Source data are provided with this paper.

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

## Acknowledgements

This research was supported by Grants-in-Aid for Scientific Research from the Japan Society for the Promotion of Science (JSPS) (Grant Nos. 25H00849 (M.H.), 22J11423 (T.I.), 22KJ0409 (T.I.), 23K22422 (M.H.), 24K01286 (T.A.), 24H00416 (S.Y.), and 23H00264 (H.S.)), and by CREST, Japan Science and Technology Agency (Grant No. JPMJCR1875) (M.H.). T.I. acknowledges the support from a Grant-in-Aid for Japan Society for the Promotion of Science (JSPS) Fellows.

## Author contributions

M.H. and T.A. planned and organized this project. S.Y. and H.S. fabricated the TMD sample. D.P. and T.A. fabricated the diamond nanoprobe. D.S., J.G. and M.H. performed experiments and analyzed the data. T.I. assisted in the measurements. All authors discussed the results. M.H., J.G., and P.F. co-wrote the manuscript.

## Competing interests

The authors declare no competing interests.
