## [Transparent Peer Review file · Nature Communications]

An ultrafast diamond nonlinear photonic sensor

Corresponding Author: Professor Muneaki Hase

Version 0:

Reviewer comments:

Reviewer #1

(Remarks to the Author)

In this article, the authors proposed a high spatial and high time resolution electric field measurement method based on femtosecond lasers and NV color center scanning probe. The authors used this method to measure the electro-optical effect of femtosecond lasers on GaAs and WSe₂ with different numbers of layers. This research is interesting, but I cannot recommend this article for publication in its current state.

I have the following concerns about this article:

1. The article barely mentions the role of NV centers here and how they are used to measure magnetic fields through second-order nonlinearity. I think this should be the most important principle of this article, which should be explained clearly through energy level diagrams in the method section. The current article does not even mention what physical quantity is measured to obtain the electric field.
2. The article mentions using AFM contact mode to let a probe containing NV centers scan the sample. The height of the contact mode on different AFMs is different on different materials. Can the author give the experimental distance between the NV center and the sample surface? This may have a great impact on the spatial resolution. From the fluorescence imaging in Figure 2, the size of the NV center ensemble should be close to 1 micron. How did the author achieve a spatial resolution of 500nm?
3. The authors mention that the experimental time resolution of ODMR is in the millisecond range. In fact, using pulsed ODMR, it is easy to achieve time-resolved measurements of tens of nanoseconds, such as high-temperature measurements based on NV centers.
4. The authors mention that NV centers are the only color centers that can be used for electro-optical sensing because they break the spatial symmetry. In fact, there are many color centers that break the symmetry of space, and those with symmetry such as SiV and GeV are only a minority.
5. Have the authors tried varying the distance between the scanning probe and the sample to measure the vertical distribution of the electric field?
6. Will the charge states of NV⁰ and NV⁻ affect the measurement results? These two charge states have different quantum efficiencies and are easily affected by electric fields.
7. This experiment is very interesting. I hope the author can provide more experimental details to help readers understand the physical process involved.

In summary, I think the research in this article is very meaningful, but the current state is not suitable for publication. I hope the author can improve it.

Reviewer #2

(Remarks to the Author)

It is well known that the commercial quantum scanning tips (See QST products in QZabre company) based on nitrogen-vacancy (NV) center in diamond always yield high spatial resolution (~10 nm). By making use of quantum coherence with time-domain quantum sensing protocol, high measurement sensitivity can be also realized in experiment. In this article, the authors use NV center to break the spatial inversion symmetry of diamond and to measure the relaxation time constants in 2D material without employing quantum coherence. After reading the manuscript very carefully, I do not recommend their work for publication in the Nature Communications journal for current results. The main reasons are as follows:

1. I can not figure out the reason for the breaking the spatial inversion symmetry of diamond with NV center to probe the electro-optic (EO) effect. To my knowledge, there are lots of semiconductor transparent materials with non-zero second-order

nonlinear susceptibility, such as SiC, GaN and LiNbO₃. They are available, easy to microfabrication and cheap compared with diamond. Why doesn't the author try these materials or compare their measurements with diamond tips?

2. From the Figure 2 d and 2 e, I can not see the diamond tip very well. And it looks like the micro-nano fabrication technology is terrible, since the diamond microstructure seems to be broken in Figure 2 e. Hence, what does the geometry of diamond tip look like? The radius of the diamond probe tip (See QST products in QZabre company) is less than 100 nm. The commercial quantum scanning tips always yield high spatial resolution (~10 nm). In the contrast, the spatial resolution in your work is too poor (~500 nm), which is same with widefield imaging. I do really think the existing results can also be achieved by traditional advanced optical methods, such as Structured Illumination Microscopy (SIM). Due to this bad spatial resolution, the authors can not provide the spatial distribution details of electron diffusion in 2D material except a mediocre average, which is same with traditional measurement. Hence, no advantage in spatial resolution is shown in this work for the use of the quantum diamond tip. Compared with traditional methods, has the detection time of diamond tips become longer or shorter under the same signal to noise ratio?

3. From Eq (1), the changing of surface electrical field is measured by light reflectivity. However, how to measure the ΔR_{eo} , R_0 in your experiment? Can the authors show the corresponding measurement optical path with all optical components? In this experiment, the time resolution of the coincidence detector of the electrical signal is very high (<1 ps) as shown in Figure 4 e, can the authors tell me which photodetector you used in experiment?

4. The number of layers of 2D materials is always measured by the Raman spectroscopy and the corresponding experimental results should be given in text. In the Figure 4 c, the fluctuation of height is too big and can the authors tell me the reason?

Reviewer #3

(Remarks to the Author)

One of the generic applications for color centers in diamond is the measurement of electric field and magnetic field. Compared with the magnetic field sensing, the electric field does not interact directly with the spin degree of freedom, but rather interacts with this degree of freedom via spin-orbital coupling. Since 2011, people have been able to measure the electric field with a reasonably good sensitivity using ODMR measurements (e.g., Dolde et al., Nature Physics (2011)). Following works discuss the Stark shift or the electric field dependence on the optical transitions (e.g. Bassett et al, PRL (2011)). Combined with advanced techniques such as scanning probe microscopy, the spatial resolution of such a method could reach sub-nm (e.g. Bian et al, Nature Communications (2021)). However, the ODMR measurement is always limited by the measurement bandwidth and making high time-resolution remains a challenge to the NV-based sensing techniques.

In this manuscript, Sato et al. proposed using the EO effect combined with the NV systems to reach a nanometer-femtosecond spatiotemporal resolution. Although both high spatial-resolution electric field sensing for NV systems (see reference above) and EO effects for measuring electric field have already been done in previous works (e.g., Ref. [5] and Ref. [6]), combining those effects to reach a high spatiotemporal resolution is still valuable and worth publication in high-profile journals.

Overall, the manuscript is intriguing and solves critical problems in the NV-based sensing. However, the current manuscript is not well-written and the representation could be very confusing for non-expert readers. With a lot of regret, I am afraid that the current version is inappropriate for the publication in Nature Communications for a broad audience. I will request the authors to answer several questions and revise the manuscript before I can recommend the publication.

Comment 1:

The manuscript lacks a general introduction to NV sensing or even an NV background, which could confuse non-expert readers. It now starts from the fabrication details for the device, which could be moved to the Methods section. Specifically, I suggest summarizing Lines 66-83 with several sentences and moving the other descriptions to the Methods section.

Comment 2:

The authors may consider adding a schematic drawing in Fig. 1, on which measurements they perform for electric field measurement. The current Fig. 1 seems to focus on the fact that SiV does not provide second-order nonlinear optical effects, which is important but not the key points illustrated in the manuscript, the description on the inverse symmetry does not need a separate figure to describe, could be just 1-2 sentences in the main text. When I read the manuscript for the first time, I was more confused but interested in how to detect the EO effect in the NVs in diamond tip but not the fact that the Group-IV vacancies could not measure it. Although the technique was described by the previous work with the same group (Ichikawa et al, Nature Communications (2024)), I suggest talking briefly about the schematics the authors will perform in Fig. 1, which could make the results more approachable to a broader audience.

Comment 3:

The Supplementary Materials need much more discussion than just two individual figures. At the very least, the authors should discuss the NV-based EO process and theory/protocol they used. It should contain a drawing for the setup and the details for the experiments. From the materials in the current manuscript (both main text and supplementary Information), it is extremely hard for the readership to understand the measurements or reproduce the results.

Comment 4:

Following my last comment, it is beneficial for the author to add a brief theoretical discussion (even ideally the full

derivations) on the sensing protocol. I will suggest starting from the Hamiltonian and having a derivation there about how $\chi^{(2)}$ plays a role here. Then it should automatically provide evidence that the protocol works for NV but not other Group-IV vacancies.

Comment 5:

The manuscript seems to miss many important citations for NV-based electric field sensing. For example, in line 43, only magnetic field papers are cited. I suggested the authors to include the reference papers for electric field sensing. For example, Dolde et al, Nature Physics (2011), Bassett et al, PRL (2011).

Comment 6:

For the background in line 41-46, the authors talk about the drawback of using ODMR for NV sensing, which was widely investigated in the magnetometry. However, the electric field only interacts indirectly with the spin, so I expect it is better to use optical transition, which has a better response (Stark shift) also a larger bandwidth. For example Bassett et al, PRL (2011). Could the author also comment on the drawbacks and a fair comparison with the method? From reading the previous papers from the same group, the protocol is also related to the optical measurement but not spin degree of freedom measurement. I can see that the EO modulation should have better temporal resolution, but it may be better for the authors to comment on it in the main text for a broad audience.

Comment 7:

Please fix typos:

Line 14: Should be "a critical task"

Line 18: Should be "photonic sensor"

Line 225: Should be "degradation"

Comment 8:

I am interested in the comparison between the NV-based technique and other EO-based electric field measurements, if any. Could the author add some information about this? Typically the NV-based magnetometer could be comparable to the state-of-the-art techniques (see Barry et al, PRApplied (2024) and Wang et al, Nature Communications (2024)). For single-NV with high resolution also could reach a fairly high sensitivity. Therefore I want to know if the NV-based electrometer could also be comparable with other techniques. This comparison will be of great interest to a broader community.

Reviewer #4

(Remarks to the Author)

In the manuscript titled 'Ultrafast diamond nonlinear photonic sensor' by Daisuke Sato et al., Authors present a concept of a nanoscale sensor of electric fields, based on reflective electro-optic response. Here, nitrogen-vacancy (NV) color centers are created in a miniature diamond to break the centrosymmetry and induce non-zero electro-optic (Pockels) effect. With such a diamond probe mounted on a scanning probe microscope, optical readout of local electric fields is proposed, with temporal resolution of 10-fs set by a pump laser pulse. Moreover, Authors propose the use of the same NV diamond for complementary magnetic and temperature measurements, albeit only in conclusions.

Overall, I have found this work novel and of high significance for studies of 2D materials. The high temporal resolution (sub-1 ps) is evident from provided figures. Similarly, the spatial resolution is around or just under 1 micron. I have only some minor remarks regarding the manuscript which I list below.

1. When reading the manuscript, I was missing a clear explanation that the method relies on reflective EO response. 2. When introducing diamond sensor preparation, a nearly 13-times enhancement is reported, but lacks explanation with respect to what? And how is the optimum NV density defined. Some explanation here is needed.
3. Equation (1) is introduced without explanation of what is ΔR_{eo} or R_0 . Also, it relates only to material properties of GaAs. Whereas that manuscript section is introduced with mentioning 'evaluation of the nonlinear optical response of the diamond NV probe'. This is confusing for the reader, as the NV-probe EO response is only inferred from Fig. 3c.
4. Further, Authors claim that 'The signal reduction of $\approx 1/42$ suggests that the probing depth of the diamond NV tip is ≤ 2.4 nm'. How is this depth inferred? How can it be justified given that mean NV centers depth in diamond is on the order of 40 nm (Ref. 18 and Ref. 45 therein) for nitrogen implantated at 30 keV. It's hard for me to imagine NV centers being sensitive only to a 2.4-nm-thin layer while being a 40 nm stand-off distance. I would like to see a comment on this.
5. In section Methods, I am missing information on pulses polarization. Is it the same arrangement as in the previous Nat. Commun. paper (Ref. 18)?
6. I am wondering if the NV sensor concept can be improved by appropriately shaping the diamond. For instance, there are commercially available (Qnami, Qzabre) AFM chips with NV diamond tips. One can already purchase a chip with a single NV center (can be either spatially oriented) or NV ensembles. Such tips can have the NV just a few nm away from the AFM tip apex, bringing it very close to the sample and increasing the spatial resolution to a few nm as well. Additionally, the diamond tip may be parabolically shaped further improving coupling of the probe beam with the NV. Would such chips be a

good candidate for the improvements suggested in Conclusions?

Version 1:

Reviewer comments:

Reviewer #1

(Remarks to the Author)

Thanks to the reviewer for his reply. After the revision, I think the article has made great progress and the physical process of the whole measurement is clearer. The measurement scheme proposed by the author is very novel. Although the spatial resolution is not as good as the existing commercial NV-AFM, this new scheme still has room for improvement in the future and is of great significance to NV sensing. I recommend this article to be published in Nature Communications.

Reviewer #2

(Remarks to the Author)

In this article, the authors proposed a high spatial and high time resolution electric field measurement method based on femtosecond lasers and diamond scanning probe. After the author modified the paper substantially, I still do not recommend their work for publication in the Nature Communications journal. There are still some comments about this work as follows:

(1) The author admits that NV center is used to break the spatial inversion symmetry of diamond to probe the electro-optic (EO) effect instead of magnetic and even temperature sensing in this work. And there are indeed lots of semiconductor transparent materials with a significant non-zero second-order nonlinear susceptibility, which can detect the electro-optic (EO) effect very accurately. Furthermore, there is a paramagnetic defect in GaN which has similar quantum sensing performance (See Luo, Jialun, et al. "Room temperature optically detected magnetic resonance of single spins in GaN." *Nat. Mater.* 23, 512-518 (2024)). The GaN materials should be used for probing the electro-optic (EO) effect, magnetic field and temperature instead of diamond materials.

(2) The experimental content of investigation WSe₂ films with diamond is too few to form a systematic and clear physical viewpoint. Transition metal dichalcogenides are a promising family of materials for electronics and optoelectronics, in part due to their range of bandgaps that can be modulated by layer number. And WSe₂ can be selectively grown with one, two, or three layers, as regulated by a one-step hydrogen-controlled chemical vapor deposition (H-CVD) process involving cyclical pulses of H₂ flow (See DeGregorio, Zachary P., Jason C. Myers, and Stephen A. Campbell. "Rational control of WSe₂ layer number via hydrogen-controlled chemical vapor deposition." *Nanotechnology* 31, 315604(2020).). The electro-optic (EO) effect of the resulting mono-, bi-, and tri-layer WSe₂ films should be investigated in this work. Furthermore, I think the substrate have an impact on the physical properties of WSe₂ films and different substrates should be investigated in experiment instead of Si substrate only.

(3) In the supplementary note 4, the density of NV- centers in diamond can be enhanced by high energy electron beam irradiation (See Capelli, Marco, et al. "Increased nitrogen-vacancy centre creation yield in diamond through electron beam irradiation at high temperature." *Carbon* 143, 714-719(2019)). Hence, I think the intensity of EO signal will be improved with this method and should be investigated in the experiment.

(4) Since the quantum coherence of NV center is not employed in this work (Degen, Christian L., Friedemann Reinhard, and Paola Cappellaro. "Quantum sensing." *Rev. Mod. Phys.* 89, 035002(2017).), the terminology of quantum sensing or quantum device based on NV center in the diamond should be deleted or modified to get rid of misleading during academic exchanges.

Reviewer #3

(Remarks to the Author)

The authors have made dramatic revisions to both my comments and those of other reviewers. I am satisfied with all the replies for the scientific arguments. As I mentioned in the previous review, with proper revisions, this work will have a significant impact on a broad audience. However, after re-reading the manuscript carefully, I still have several minor suggestions. I am hesitant about whether another round of review will be needed before formally recommending the publication, but at the very least, I strongly encourage the authors to further revise the manuscript to improve clarity and maximize its potential to attract interest and citations.

1. There are still several typos. For example, in line 83, it should read "Diamond nonlinear photonic sensor" rather than "photonics sensor." Additionally, the acronym SIM is defined inconsistently: it stands for Structured Illumination Microscopy on line 65, but refers to Scanning Ion Microscopy on line 92. The authors should revise the text to resolve this inconsistency.

2. In line 46, I do not believe that "quantum emitters" constitute an "application" on the same level as quantum computing and quantum communication. I suggest using the term "quantum networks" instead, as it more accurately represents one of the current state-of-the-art applications of quantum emitters. Personally, I would not recommend placing emphasis on the potential application of NV centers in quantum computing, as there are still many unresolved challenges. Moreover, there

are already numerous misconceptions and misleading terms in this area, which have caused considerable confusion among audiences. That said, this is merely my personal opinion, and the final decision ultimately rests with the authors.

3. Following my last review, I have suggestions for the authors regarding the paper citations, as several citations still do not align with the contents of the author's manuscript.

For example, Ref. 16 describes the high-fidelity transfer and storage of information using nuclear spins, which does not necessarily pertain directly to quantum communication. Several other works may be more relevant in this context. For instance, Pompili et al., *Science* (2021), and Knaut et al., *Nature* (2024), report on the development of a multimode quantum internet using remote NV and SiV centers. Additionally, Wang et al., *Nature Communications* 14, 704 (2023), present advancements in quantum repeater and quantum router architectures for remote quantum communication. I suggest that the authors consider citing these more pertinent works.

The authors mentioned group-IV vacancies but also do not add appropriate citations. Ref. 26 does not show the inversion symmetry etc. A much more appropriate citation will be Hepp et al, *PRL* (2014) for SiV.

I suggest the authors to check all the referred papers again.

4. What are BV, OV centers on line 64? There is no definition on this.

5. I think line 73 may be more suitable for Methods section, like AFM pin point mode may be too detailed in the main text, especially in the introduction session.

6. The introduction session is not well-written. The connection between the second and third paragraph is vague. What are the points to mention other quantum defects in line 61-63 and why mentioning the reason not to use other EO crystals? Those discussions are more suitable to the conclusion but not introduction. The readers will not care what are not describing in the manuscript in the introductions. The current version is misleading and could be improved in a further revision.

7. To reiterate, I have never questioned the scientific value of the manuscript. However, I strongly recommend that the authors carefully revise the paper once more and correct the remaining typographical errors before publication. Since it will potentially be published in a prestigious journal like *Nature Communications*, it is important to maintain a high standard of clarity, conciseness, and overall writing quality.

Reviewer #4

(Remarks to the Author)

In the amended manuscript, Authors have addressed points raised in my review, as well as over Reviewers' ones. In particular, they explained now much better the principles of measuring the EO response. I am content with the changes made and sure these make the manuscript easier to read.

The only left over remark I have is the missing discussion of the implantation depth and thickness of the NV-layer created and used for the experiment.

For non-expert readers, it may be difficult to assess how the NV-layer looks like if it's characterized only by the concentration or ion fluence. Moreover, the sentence: "Thus, the height of the NV center probe was quasi-zero at each of the designated 76 points, which was determined by the Lennard-Jones potential (Typically 0.1-0.3 nm)" is misleading since it discusses the distance of the diamond (plate) rather than NVs (layer in that plate) from the surface.

Apart from this issue, I believe the article is appropriate for publication in *Nature Communications*, as it brings attention to a novel and interesting application of NV diamonds.

Response to Reviewer #1

We would like to express our gratitude to the reviewers for reading our manuscript and providing fruitful comments. We have conducted additional experiments after receiving the initial decision of the editors and have revised our manuscript following the reviewers' suggestions.

[Referee]

In this article, the authors proposed a high spatial and high time resolution electric field measurement method based on femtosecond lasers and NV color center scanning probe. The authors used this method to measure the electro-optical effect of femtosecond lasers on GaAs and WSe₂ with different numbers of layers. This research is interesting, but I cannot recommend this article for publication in its current state.

I have the following concerns about this article:

(1) The article barely mentions the role of NV centers here and how they are used to measure magnetic fields through second-order nonlinearity. I think this should be the most important principle of this article, which should be explained clearly through energy level diagrams in the method section. The current article does not even mention what physical quantity is measured to obtain the electric field.

#Our reply#

The measurement of the electric field is based on the linear electro-optic effect (i.e., the Pockels effect; the linear variation of the optical index of refraction upon application of quasi-static electric field), which is a second-order nonlinear optical effect. Since the electro-optic effect is basically induced by non-resonant transitions, it does not require a real charge carrier excitation. That means our 1.5 eV photon ($\hbar\omega$) does not directly interact with the NV- spin states as schematically shown in Fig. R1.

Fig. R1. The energy level diagram around the NV- center in diamond. The 1.5 eV photon ($\hbar\omega$) used in the present study does not interact with the NV- spin states directly, but non-resonant transitions occur from the ground state to/from a virtual state, corresponding to the Pockels effect; $\chi_{ij}^{(2)}(\omega = \omega - 0)$.

We now explain what physical quantity is measured to obtain the electric field by means of the nonlinear optical process rather than the conventional ODMR method. The refractive index under quasi-static electric field E accompanying the Pockels and Kerr electro-effect is given by,

$$n(E) = n_0 - \frac{1}{2}n_0^3r^{ij}E + n_2E^2 \quad (\text{R1})$$

Where n_0 is the static refractive index, r^{ij} and n_2 are the Pockels and Kerr coefficient, respectively. Using the general relations of $n_2 = \frac{3}{4n_0^2\epsilon_0 c} \chi_{ijk}^{(3)}$ and $r^{ij} = -\frac{2}{n_0^4} \chi_{ij}^{(2)}$ (please refer e.g., R. W. Boyd, *Nonlinear Optics*, Academic Press, 2008) we have

$$n(E) = n_0 + \frac{1}{n_0} \chi_{ij}^{(2)} E + \frac{3}{4n_0^2\epsilon_0 c} \chi_{ijk}^{(3)} E^2 \quad (\text{R2})$$

In the case of a NV center introduced in diamond, in which inversion symmetry is broken, we have $\chi_{ij}^{(2)} \neq 0$ and $\chi_{ijk}^{(3)} \neq 0$, so $\Delta n(E)$ is the sum of the two terms, whereas in the case of other quantum defects, such as the SiV center, $\chi_{ij}^{(2)} = 0$ and $\chi_{ijk}^{(3)} \neq 0$, so $n(E)$ is given only by the third-order term, which is generally smaller than the second-order term. The 13-fold enhancement of the EO response observed in NV diamond can thus be explained by the additional second-order term because of $\chi_{ij}^{(2)} \neq 0$.

Fig. R2. (a) Electro-optic detection of an anisotropic change in the refractive index. The polarization of the probe was 45° with respect to the optical plane. (b) The isotropic refractive index (n_0) before photoexcitation ($E=0$) and the anisotropic refractive index (n_{eo}) after photoexcitation ($E \neq 0$).

In the experiment, we measure the anisotropic reflectivity change $\Delta R_{eo}(t)$ as the EO signal,

$$\frac{\Delta R_{eo}(t)}{R_0} = \frac{4}{n_0^2 - 1} (\Delta n_x(t) - \Delta n_y(t)) = \frac{4n_0^3}{(n_0^2 - 1)} r^{ij} \Delta E(t) = \frac{-8}{(n_0^2 - 1)n_0} \chi_{ij}^{(2)} \Delta E(t) \quad (\text{R3})$$

where $n_x(t) = n_0 + \frac{1}{2} n_0^3 r^{ij} E$ and $n_y(t) = n_0 - \frac{1}{2} n_0^3 r^{ij} E$ are the x- and y-components of the indices of refraction, and R_0 is the reflectivity without photoexcitation, as shown in Fig. R2. Here, we omit the third-order term since it is orders of magnitude smaller than the second-order term. Eq. (R3) clearly indicates the EO sampling is a measure of the change of the electric field $\Delta E(t)$ and this is only possible when $\chi_{ij}^{(2)} \neq 0$, corresponding to the case of NV diamond. In the revision, we have added these notes both in the Methods section and the Supplementary Information.

[Referee]

(2) The article mentions using AFM contact mode to let a probe containing NV centers scan the sample. The height of the contact mode on different AFMs is different on different materials. Can the author give the experimental distance between the NV center and the sample surface? This may have a great impact on the spatial resolution. From the fluorescence imaging in Figure 2, the size of the NV center ensemble should be close to 1 micron. How did the author achieve a spatial resolution of 500nm?

#Our reply#

Thank you for the important question about the contact mode measurements. Although we mentioned using AFM contact mode, it was used only when taking the topography of the sample (Fig. 4a). For the time-resolved EO measurements presented in Figure 3c and 4e, we have in fact used the “pin-point mode”, that is vertically approaching and retracting the NV probe at each designated points on the sample as shown in Fig. R3. Thus, the height of the NV center probe was quasi-zero at each of the designated points (P4-P11 in Fig. 4e), which is determined by the Lennard-Jones potentials (Typically 0.1-0.3 nm).

Fig. R3. The schematic of the “pin-point mode”, that is vertically approaching and retracting the AFM probe at each designated points on the sample.

To investigate if our NV tip really approached on sample surface (the height was quasi-zero), we took a force-distance curve for an n-GaAs wafer sample in air, as shown in Fig. R4. When using a commercial Si-tip from Sensor Tech Inc (PRSA-L300-F50-Si-PCB; Tip radius < 15 nm) the force-distance curve shows a snap into contact at ~100 nm and approached at T-B (resistance bridge; top minus bottom) = 35 mV, while in the case using the NV-tip it shows a snap into contact at ~180 nm and approached at T-B = 35 mV and when retracted it shows a jump out at ~ 420 nm. Thus, we can conclude that the NV tip really approached the sample surface.

Fig. R4. (a) The force curve measured for Si-tip on n-GaAs wafer. (b) The force curve measured for NV-tip on n-GaAs wafer. In both plots, T-B=35 mV corresponds to the approached point.

Lastly, from the fluorescence imaging shown in Fig. 2b, the size of the NV center ensemble was estimated as shown in Fig. R5. Based on a Gaussian fit of the line profile near the center, the spatial resolution of the diamond NV probe was found to be better than ≈ 660 nm and even potentially ≤ 500 nm because of enhancement of the EO sensitivity at the apex of the NV tip: the $^{14}\text{N}^+$ ion dose at the most intense red-color region ≈ 500 nm in Fig. 2b is the same as the optimal density of $1 \times 10^{12} \text{ cm}^{-2}$ found in the previous study [Ichikawa *et al.*, Nat. Commun. **15**, 7174 (2024)]. In the revision, we have added these notes in the main text and in the Supplementary Information.

Fig. R5. The photoluminescence (PL) image (Top) and the line profile (Bottom) at the horizontal dashed line. Based on the Gaussian fit of the line profile, the spatial resolution of the diamond NV probe was better than ≈ 660 nm and potentially ≤ 500 nm due to enhancement of the EO sensitivity at the apex of the NV tip.

[Referee]

(3) The authors mention that the experimental time resolution of ODMR is in the millisecond range. In fact, using pulsed ODMR, it is easy to achieve time-resolved measurements of tens of nanoseconds, such as high-temperature measurements based on NV centers.

#Our reply#

Thank you for the comment on the time resolution of ODMR. We agree with the reviewer's statement, and we have corrected "millisecond range" into "nanosecond range" in the revised manuscript. We have also added a few related reference papers, D. M. Toyli *et al*, Phys. Rev. X, Vol.2, 031001 (2012); J.-. W. Fan *et al*, Nano Lett. Vol.24, 14806 (2024).

[Referee]

(4) The authors mention that NV centers are the only color centers that can be used for electro-optical sensing because they break the spatial symmetry. In fact, there are many color centers that break the symmetry of space, and those with symmetry such as SiV and GeV are only a minority.

#Our reply#

Thank you for pointing out the issue regarding our comment on color centers that break the symmetry of space. In the revision, we have appropriately mentioned that other color centers, such as BV and OV centers [T. Umeda *et al*, Phys. Rev. B, Vol.105, 165201 (2022)], can also break spatial symmetry.

[Referee]

(5) Have the authors tried varying the distance between the scanning probe and the sample to measure the vertical distribution of the electric field?

#Our reply#

Thank you for the interesting suggestion. We have not yet tried varying the distance between the scanning probe and the sample to measure the vertical distribution of the electric field. From the force curve measurements shown in Fig. R4, we found the NV tip has larger contact region up to

~ 420 nm. So, we probably need to try to make a smaller apex tip or build a new AFM system which can be used in a vacuum. We do plan to conduct such measurements but they will require additional time and budget. In addition, “truly non-contact mode”, i.e., an oscillating tip that maintains a fixed distance between the tip and sample, might also prove useful. In this case, it will be required to build a new protocol to update our current system. In the revision, we note these ideas as future prospective experiments.

[Referee]

(6) Will the charge states of NV⁰ and NV⁻ affect the measurement results? These two charge states have different quantum efficiencies and are easily affected by electric fields.

#Our reply#

Thank you for the interesting question. We believe that the charge state of the NV⁻ center may play a central role in the strong enhancement of the EO effect as presented in our previous work, Ichikawa *et al.*, Nat. Commun. **15**, 7174 (2024); Ref. [18]. In the supplementary file of Ref. [18], we showed the density of NV⁻ state was a maximum at a N⁺ ion dose of $1 \times 10^{12} \text{ cm}^{-2}$ based on ODMR measurements, which is shown again in Fig. R6 below. Thus, the ODMR results indicate the optimal density for enhancement of the EO effect is coincident with the maximum NV⁻ density. In the revision, we have added the statement that the charge state of NV⁻ may play a central role in the EO response.

Fig. R6. Optically detected magnetic resonance (ODMR) spectra obtained at room temperature for NV diamond samples using 532 nm cw laser. The ODMR measurements show the contrast of the NV⁻ resonant dip was maximized (14.5 %) at the dose of $1 \times 10^{12} \text{ N}^+ \text{ cm}^{-2}$, indicating the density of NV⁻ was enhanced at $1 \times 10^{12} \text{ N}^+ \text{ cm}^{-2}$.

[Referee]

(7) This experiment is very interesting. I hope the author can provide more experimental details to help readers understand the physical process involved.

In summary, I think the research in this article is very meaningful, but the current state is not suitable for publication. I hope the author can improve it.

#Our reply#

Thank you again for carefully reading the manuscript and offering fruitful comments. After addressing all of comments, we hope the revised version is easier to follow.

Response to Reviewer #2:

We are grateful to the reviewers for reading our manuscript and their fruitful comments. We have conducted additional experiments after receiving the initial editor decision and now have revised the manuscript following the reviewer's suggestions.

[Referee]

It is well known that the commercial quantum scanning tips (See QST products in QZabre company) based on nitrogen-vacancy (NV) center in diamond always yield high spatial resolution (~10 nm). By making use of quantum coherence with time-domain quantum sensing protocol, high measurement sensitivity can be also realized in experiment. In this article, the authors use NV center to break the spatial inversion symmetry of diamond and to measure the relaxation time constants in 2D material without employing quantum coherence. After reading the manuscript very carefully, I do not recommend their work for publication in the Nature Communications journal for current results. The main reasons are as follows:

1. I can not figure out the reason for the breaking the spatial inversion symmetry of diamond with NV center to probe the electro-optic (EO) effect. To my knowledge, there are lots of semiconductor transparent materials with non-zero second-order nonlinear susceptibility, such as SiC, GaN and LiNbO₃. They are available, easy to microfabrication and cheap compared with diamond. Why doesn't the author try these materials or compare their measurements with diamond tips?

#Our reply#

Thank you for the suggestion of possible semiconductor materials other than NV diamond. We have in fact tried to fabricate a tip with semiconductor GaP wafers, which is also one of a EO crystal (Pockels coefficient of $r_{41} \approx -1.0 \text{ pm/V}$). Although GaP would be easy for microfabrication and less expensive compared with diamond, it would still take time to fabricate a useful GaP tip, and therefore, we plan a comparison of such measurements in a potential future work.

The main reason for using the diamond tip with NV center is a possibility of magnetic and even temperature sensing, which would not be possible using another EO crystal, such as GaP, 3C-SiC (Pockels coefficient of $r_{33} \approx 2.7 \text{ pm/V}$), GaN (Pockels coefficient of $r_{33} \approx 1.9 \text{ pm/V}$) and LiNbO₃ (Pockels coefficient of $r_{33} \approx 31 \text{ pm/V}$). In addition, the spatial resolution could potentially be <10 nm if the sensitivity could be increased down to only several or even a single NV level. In the revision, we have added these comments.

[Referee]

2. From the Figure 2 d and 2 e, I can not see the diamond tip very well. And it looks like the micro-nano fabrication technology is terrible, since the diamond microstructure seems to be broken in Figure 2 e. Hence, what does the geometry of diamond tip look like? The radius of the diamond probe tip (See QST products in QZabre company) is less than 100 nm. The commercial quantum scanning tips always yield high spatial resolution (~10 nm). In the contrast, the spatial resolution in your work is too poor (~500 nm), which is same with widefield imaging. I do really think the existing results can also be achieved by traditional advanced optical methods, such as Structured Illumination Microscopy (SIM). Due to this bad spatial resolution, the authors can not provide the spatial distribution details of electron diffusion in 2D material except a mediocre average, which is same with traditional measurement. Hence, no advantage in spatial resolution is shown in this work for the use of

the quantum diamond tip. Compared with traditional methods, has the detection time of diamond tips become longer or shorter under the same signal to noise ratio?

#Our reply#

The current 1st AFM system is based on a self-sensing cantilever system from SENSOR Tech Inc. This was our choice when we decided upon the order of a custom-made AFM system from Anfatech Inc five years ago. We have already asked if the commercial tip (QST products in QZabre company) can be attached to the 1st AFM system, however, it looks difficult to attach because of the reduced area of our self-sensing cantilever compared to their commercial tip. Therefore, we are planning to build 2nd AFM system based on a tuning fork system, to which commercial (QST products in QZabre company) diamond NV tip can be potentially attached, although these are future works after we get a new research budget.

Structured Illumination Microscopy (SIM) is a kind of commercially available super-resolution microscopy, and the resolution is sub-micrometer. Although time-resolved SIM has also been reported, e.g., in Rodermund *et al.*, Science **372**, 1167 (2021), the time resolution is larger than seconds. So, there is still strong motivation to develop a new technique with ultrafast time resolution.

The main idea demonstrated in our manuscript is compatibility of high spatiotemporal resolution, i.e., ≤ 500 nm and ≤ 100 fs, which is still valuable in the relevant field as the other reviewers evaluated. To compare the detection time of diamond tips with traditional SIM techniques, we present just test data of EO imaging for an Au electrode on a *n*-GaAs wafer as shown in Fig. R7. The detection time here was <5 min (256x256 pixels, 1 sec/line \times 256 lines = 256 sec \approx 4.3 min.), which is comparable to the SIM technique. Note that the time delay after the pump pulse for this image was 0.1 ps (100 fs), which is in general difficult to achieve by time-resolved SIM techniques and indicates the potential of the ultrafast diamond nonlinear photonic sensor to future applications for electronic devices. In the revision, we have revised Fig. 2e to enlarge the diamond tip and added these points to discuss the differences between our method and the super-resolution microscopy (SIM) technique.

Fig. R7. Testing time-resolved EO imaging for an Au electrode on *n*-GaAs wafer obtained using the NV tip at room temperature. The pump power was 90 mW (the fluences of 215 $\mu\text{J}/\text{cm}^2$).

[Referee]

3. From Eq (1), the changing of surface electrical field is measured by light reflectivity. However, how to measure the ΔR_{eo} , R_0 in your experiment? Can the authors show the corresponding measurement optical path with all optical components? In this experiment, the time resolution of the coincidence detector of the electrical signal is very high (<1 ps) as shown in Figure 4 e, can the authors tell me which photodetector you used in experiment?

#Our reply#

Thank you for this important question. The photodetector we used was a 1 GHz high-speed Si-PIN photodetector (S5931) from Hamamatsu Photonics, Inc. The time resolution in this experiment is determined by the pulse width of the laser (~ 10 fs) in a pump-probe configuration.

To measure the electro-optic (or anisotropic reflectivity) $\Delta R_{eo}/R_0$ signal, the photocurrent from the two Si-PIN photodetector was subtracted and then amplified by a current amplifier as shown in Fig. R2. In the experiment, we measured the anisotropic reflectivity change $\Delta R_{eo}(t)$ as the EO signal,

$$\frac{\Delta R_{eo}(t)}{R_0} = \frac{4}{n_0^2 - 1} (\Delta n_x(t) - \Delta n_y(t)) = \frac{4n_0^3}{(n_0^2 - 1)} r^{ij} \Delta E(t) = \frac{-8}{(n_0^2 - 1)n_0} \chi_{ij}^{(2)} \Delta E(t) \quad (\text{R3})$$

where $n_x(t) = n_0 + \frac{1}{2}n_0^3 r^{ij} E$ and $n_y(t) = n_0 - \frac{1}{2}n_0^3 r^{ij} E$ are the x- and y-components of the indices of refraction, and R_0 is the reflectivity without photoexcitation. Eq. (R3) clearly indicates the EO sampling is the measure of the change in the electric field $\Delta E(t)$ and this is only possible when $\chi_{ij}^{(2)} \neq 0$, corresponding to the case of the NV diamond.

Fig. R2. (a) Electro-optic detection of the anisotropic change in the refractive index. The polarization of the probe was 45° with respect to the optical plane. (b) Isotropic refractive index (n_0) before the photoexcitation ($E=0$) and the anisotropic refractive index (n_{eo}) after the photoexcitation ($E \neq 0$).

The optical path of all optical components for the $\Delta R_{eo}/R_0$ measurement is shown in Fig. R8 and is a well-established experimental technique in the field of ultrafast laser spectroscopy [Please refer e.g., G. C. Cho *et al*, Phys. Rev. Lett. Vol. **65**, 764–766 (1990); M. Hase *et al*, Nature Vol. **426**, 51–54 (2003); M. Hase *et al*, Nature Photon. Vol. **6**, 243 (2012)]. We have built a homemade microscopy system combined with a self-sensing AFM system, as schematically shown in Fig. R9 and as a photograph in Fig. R10. In the revision, we have added these notes in the Supplementary Information.

Fig. R8. Time-resolved pump-probe electro-optic (EO) sampling setup using a 10-fs pulsed laser. The time delay between the pump and probe pulses was scanned using an oscillating mirror. Both pump and probe beams were focused onto the sample by a parabolic mirror, and only reflected probe was detected and amplified to obtain the $\Delta R_{eo}/R_0$ signal.

Fig. R9. Schematic of the AFM system designed on a breadboard. (Left) Side view. (Right) Top view.

Fig. R10. (Left) Photograph of our AFM system, designed in Fig. R9. (Right) The enlarged photo of sample holder and piezo scanner under the illumination of both pump and probe beams.

[Referee]

4. The number of layers of 2D materials is always measured by the Raman spectroscopy and the corresponding experimental results should be given in text. In the Figure 4 c, the fluctuation of height is too big and can the authors tell me the reason?

#Our reply#

Thank you for the important question. The Raman spectra taken for 1ML and Bulk-WSe₂ are shown in Fig. R11 below. The 1ML-WSe₂ shows strong peak at 245 cm⁻¹ together with a satellite peak at 254 cm⁻¹, whereas Bulk-WSe₂ shows weak signals at 245 cm⁻¹ and 252 cm⁻¹, together with a small peak at 306 cm⁻¹, which indicate Raman signature as expected for 1ML and Bulk-WSe₂ (M. De Luca *et al.*, 2D Materials Vol.7 025004 (2020); P. Tonndorf *et al.*, Opt. Exp. Vol.21, 4908 (2013); W. Zhao *et al.*, Nanoscale, Vol.5, 9677 (2013); H. Terrones *et al.*, Sci. Rep. Vol.4, 4215 (2014); E. del Corro *et al.*, ACS Nano, Vol.8, 9629 (2014)), although a small redshift in the peak positions might originate from the effects of transfer process of WSe₂, e.g., carrier doping, surface oxidation, defects formation, and stress formation. In the revision, we have added information of the Raman spectra in the Supplementary Information.

Fig. R11. Raman spectra obtained for 1ML-WSe₂ and Bulk-WSe₂ using a 532 nm laser at room temperature.

Lastly, although the fluctuation of height are a concern in Figure 4c; possible reasons are surface wrinkles of the WSe₂ sample and vibrations from the clean booth system, in which we have placed the optical table and laser systems. We have added these notes in the figure caption of Figure 4c in the revised version.

Response to Reviewer #3:

We are grateful to the reviewers for reading our manuscript and their fruitful comments. We have conducted additional experiments after receiving the initial editor decision and now have revised the manuscript following the reviewer's suggestions.

[Referee]

One of the generic applications for color centers in diamond is the measurement of electric field and magnetic field. Compared with the magnetic field sensing, the electric field does not interact directly with the spin degree of freedom, but rather interacts with this degree of freedom via spin-orbital coupling. Since 2011, people have been able to measure the electric field with a reasonably good sensitivity using ODMR measurements (e.g., Dolde et al., Nature Physics (2011)). Following works discuss the Stark shift or the electric field dependence on the optical transitions (e.g. Bassett et al, PRL (2011)). Combined with advanced techniques such as scanning probe microscopy, the spatial resolution of such a method could reach sub-nm (e.g. Bian et al, Nature Communications (2021)). However, the ODMR measurement is always limited by the measurement bandwidth and making high time-resolution remains a challenge to the NV-based sensing techniques.

In this manuscript, Sato et al. proposed using the EO effect combined with the NV systems to reach a nanometer-femtosecond spatiotemporal resolution. Although both high spatial-resolution electric field sensing for NV systems (see reference above) and EO effects for measuring electric field have already been done in previous works (e.g., Ref. [5] and Ref. [6]), combining those effects to reach a high spatiotemporal resolution is still valuable and worth publication in high-profile journals.

Overall, the manuscript is intriguing and solves critical problems in the NV-based sensing. However, the current manuscript is not well-written and the representation could be very confusing for non-expert readers. With a lot of regret, I am afraid that the current version is inappropriate for the publication in Nature Communications for a broad audience. I will request the authors to answer several questions and revise the manuscript before I can recommend the publication.

1. The manuscript lacks a general introduction to NV sensing or even an NV background, which could confuse non-expert readers. It now starts from the fabrication details for the device, which could be moved to the Methods section. Specifically, I suggest summarizing Lines 66-83 with several sentences and moving the other descriptions to the Methods section.

#Our reply#

Thank you for the suggestion for improving the introduction. We have summarized Lines 66-83 into several sentences and moved the other detailed descriptions to the Methods section in the revised manuscript, as the reviewer suggested. Instead, we have added a general introduction to quantum technologies based on NV centers in Lines 43-46.

[Referee]

2. The authors may consider adding a schematic drawing in Fig. 1, on which measurements they perform for electric field measurement. The current Fig. 1 seems to focus on the fact that SiV does not provide second-order nonlinear optical effects, which is important but not the key points illustrated in the manuscript, the description on the inverse symmetry does not need a separate figure to describe, could be just 1-2 sentences in the main text. When I read the manuscript for the first time, I was more confused but interested in how to detect the EO

effect in the NVs in diamond tip but not the fact that the Group-IV vacancies could not measure it. Although the technique was described by the previous work with the same group (Ichikawa et al, Nature Communications (2024)), I suggest talking briefly about the schematics the authors will perform in Fig. 1, which could make the results more approachable to a broader audience.

#Our reply#

Thank you for the suggestions. Other reviewers also inquired how to detect the EO effect from the NVs in the diamond tip. To make this point clear, we have revised Fig. 1, as shown in Fig. R12 below. We have moved the current Fig. 1a (the description of the inverse symmetry) into the Supplementary Information and commented on it with just 1-2 sentences in the main text, as the reviewer suggested.

Fig. R12. (a) The schematic of the “pin-point mode”, that is vertically approaching and retracting the AFM probe at each designated points on the sample. (b) A schematic of electro-optic sampling using a diamond NV tip. The Pockels effect occurs where the spatial inversion symmetry is broken by the NV centers ($\chi^{(2)} \neq 0$), and the refractive index change ($\Delta n \propto \chi_{ij}^{(2)} \Delta E(t)$) of this part is probed. The electric field, which is detected by the NV probe, is generated by the photoexcitation of electrons at the sample surface.

[Referee]

3. The Supplementary Materials need much more discussion than just two individual figures. At the very least, the authors should discuss the NV-based EO process and theory/protocol they used. It should contain a drawing for the setup and the details for the experiments. From the materials in the current manuscript (both main text and supplementary Information), it is extremely hard for the readership to understand the measurements or reproduce the results.

#Our reply#

Thank you for suggestion for improving the Supplementary Materials, which is also pointed out by other reviewers.

To measure the electro-optic (or anisotropic reflectivity) $\Delta R_{eo}/R_0$ signal, the photocurrent from the two Si-PIN photodetector was subtracted and then amplified by a current amplifier as shown in Fig. R2. In the experiment, we measure the anisotropic reflectivity change $\Delta R_{eo}(t)$ as the EO signal,

$$\frac{\Delta R_{eo}(t)}{R_0} = \frac{4}{n_0^2 - 1} (\Delta n_x(t) - \Delta n_y(t)) = \frac{4n_0^3}{(n_0^2 - 1)} r^{ij} \Delta E(t) = \frac{-8}{(n_0^2 - 1)n_0} \chi_{ij}^{(2)} \Delta E(t) \quad (R3)$$

where $n_x(t) = n_0 + \frac{1}{2}n_0^3 r^{ij} E$ and $n_y(t) = n_0 - \frac{1}{2}n_0^3 r^{ij} E$ are the x- and y-components of the indices of refraction, and R_0 is the reflectivity without photoexcitation. Eq. (R3) clearly indicates the EO sampling is measuring the change of the electric field $\Delta E(t)$ and this is only possible when $\chi_{ij}^{(2)} \neq 0$, corresponding to the case of NV diamond.

Fig. R2. (a) Electro-optic detection of an anisotropic change in the refractive index. The polarization of the probe was 45° with respect to the optical plane. (b) Isotropic refractive index (n_0) before the photoexcitation ($E=0$) and the anisotropic refractive index (n_{eo}) after the photoexcitation ($E \neq 0$).

The optical path of all optical components for the $\Delta R_{eo}/R_0$ measurement is shown in Fig. R8, which is well established experimental technique in the field of ultrafast laser spectroscopy [Please refer e.g., G. C. Cho *et al*, Phys. Rev. Lett. Vol.65, 764–766 (1990); M. Hase *et al*, Nature Vol.426, 51-54 (2003); M. Hase *et al*, Nature Photon. Vol.6, 243 (2012)]. We have built a homemade microscopy system combined with a self-sensing AFM system, as schematically shown in Fig. R9 and as a photograph in Fig. R10. In the revised version, we have added these notes in the Supplementary Information.

Fig. R8. Time-resolved pump-probe electro-optic (EO) sampling setup using a 10-fs pulsed laser. The time delay between the pump and probe pulses was scanned by using an oscillating mirror. Both pump and probe beams were focused onto the sample by a parabolic mirror, and only reflected probe was detected and amplified to obtain the $\Delta R_{eo}/R_0$ signal.

Fig. R9. Schematic of the AFM system designed on the breadboard. (Left) Side view. (Right) Top view.

Fig. R10. (Left) Photograph of our AFM system, designed in Fig. R9. (Right) The enlarged photo of sample holder and piezo scanner under the illumination of both pump and probe beams.

[Referee]

4. Following my last comment, it is beneficial for the author to add a brief theoretical discussion (even ideally the full derivations) on the sensing protocol. I will suggest starting from the Hamiltonian and having a derivation there about how $\chi^{(2)}$ plays a role here. Then it should automatically provide evidence that the protocol works for NV but not other Group-IV vacancies.

#Our reply#

Thank you for this important question. The measurement of electric field is based on the linear electro-optic effect (i.e., Pockels effect; the linear variation of the optical index of refraction upon application of quasi-static electric field), which is a second-order nonlinear optical effect. Since the electro-optic effect is basically induced by non-resonant transitions, it does not require real charge carrier excitation. That means our 1.5 eV photon ($\hbar\omega$) does not directly interact with the NV- spin states as schematically shown in Fig. R1.

Fig. R1. The energy level diagram around the NV⁻ center in diamond. The 1.5 eV photon ($\hbar\omega$) used in the present study does not interact with the NV⁻ spin states directly, but non-resonant transitions occur from the ground state to/from the virtual state, corresponding to the Pockels effect; $\chi_{ij}^{(2)}(\omega = \omega - 0)$.

We now explain what physical quantity is measured to obtain the electric field by means of nonlinear optical process rather than the conventional ODMR method. The refractive index under a quasi-static electric field E accompanying the Pockels and Kerr electro-effect is given by,

$$n(E) = n_0 - \frac{1}{2}n_0^3 r^{ij} E + n_2 E^2 \quad (R1)$$

Where n_0 is the static refractive index, r^{ij} and n_2 are the Pockels and Kerr coefficients, respectively. Using the general relations of $n_2 = \frac{3}{4n_0^2 \epsilon_0 c} \chi_{ijk}^{(3)}$ and $r^{ij} = -\frac{2}{n_0^4} \chi_{ij}^{(2)}$ (please refer e.g., R. W. Boyd, *Nonlinear Optics*, Academic Press, 2008) we have

$$n(E) = n_0 + \frac{1}{n_0} \chi_{ij}^{(2)} E + \frac{3}{4n_0^2 \epsilon_0 c} \chi_{ijk}^{(3)} E^2 \quad (R2)$$

In the case of the NV center introduced diamond, in which inversion symmetry is broken, we have $\chi_{ij}^{(2)} \neq 0$ and $\chi_{ijk}^{(3)} \neq 0$, so $\Delta n(E)$ is the sum of the two terms, whereas in the case of other quantum defects, such as SiV centers, $\chi_{ij}^{(2)} = 0$ and $\chi_{ijk}^{(3)} \neq 0$, so $n(E)$ is given only by the third-order term, which is generally smaller than the second-order term. The 13-fold enhancement of the EO response observed in the NV diamond can thus be explained by the additional second-order term because of $\chi_{ij}^{(2)} \neq 0$. In the revision, we have added these notes into the Supplementary Information.

[Referee]

5. The manuscript seems to miss many important citations for NV-based electric field sensing. For example, in line 43, only magnetic field papers are cited. I suggested the authors to include the reference papers for electric field sensing. For example, Dolde et al, Nature Physics (2011), Bassett et al, PRL (2011).

#Our reply#

Thank you for the fruitful suggestion. We have added a few references the reviewer suggested.

[Referee]

6. For the background in line 41-46, the authors talk about the drawback of using ODMR for NV sensing, which was widely investigated in the magnetometry. However, the electric field only interacts indirectly with the spin, so I expect it is better to use optical transition, which has a better response (Stark shift) also a larger bandwidth. For example Bassett et al, PRL (2011). Could the author also comment on the drawbacks and a fair comparison with the method? From reading the previous papers from the same group, the protocol is also related to the optical measurement but not spin degree of freedom measurement. I can see that the EO modulation should have better temporal resolution, but it may be better for the authors to comment on it in the main text for a broad audience.

#Our reply#

Thank you for suggesting comparison with the Stark shift, in which the applied electric fields perturb both the ground-state spin and excited-state orbitals. The dc Stark perturbation to the Hamiltonian is,

$$H_{Stark} = -\boldsymbol{\mu} \cdot \boldsymbol{F}. \quad (\text{R4})$$

where \boldsymbol{F} is the local electric field and $\boldsymbol{\mu}$ is the electric dipole operator. Based on Bassett *et al*, PRL (2011), the transition energy eigenvalue is $E_{\pm} = h\bar{\nu} \pm \frac{1}{2}h\delta$, where $h\bar{\nu} = \hbar\omega_0 + \Delta\mu_{\parallel}F_z$ and $h\delta = \sqrt{2}(V_{E_1}^2 + V_{E_2}^2)^{1/2}$. These indicate an electric field F_z applied along the NV- center symmetry axis shifts the energy of both transitions ($A_2 \rightarrow E_1$ and $A_2 \rightarrow E_2$) by the same amount ($\Delta\mu_{\parallel}F_z$). Thus, the dc Stark effect can detect photoluminescence spectra under resonant transitions ($A_2 \rightarrow E_1$ and $A_2 \rightarrow E_2$).

In our experiment, on the contrary, the laser energy (1.55 eV) was not resonant with both transitions ($A_2 \rightarrow E_1$ and $A_2 \rightarrow E_2$) as demonstrated in Fig. R1, and therefore is not related to the conventional dc Stark effect. However, our nonlinear optical method may be related to the “optical” Stark effect, known as inverse Faraday effect (IFE) or inverse Cotton-Mouton effect (ICME), in which light-induced dc magnetization arises because the optical field shifts the different magnetic states of the ground manifold differently and mixes into these ground states different amounts of excited state (please refer page 61 of Shen, Y. R. Principles of Nonlinear Optics. *Wiley-Interscience, New York* (1984)). We have in fact observed both IFE and ICME in NV diamond in Ref.[43] (Sakurai et al., APL Photon. **7**, 066105 (2022)). We are greatly interested in further studying the “optical” Stark effect in the future. In the revision, we have added comparison of our method with dc Stark effect appropriately.

[Referee]

7. Please fix typos:

Line 14: Should be “a critical task”

Line 18: Should be “photonic sensor”

Line 225: Should be “degradation”

#Our reply#

Thank you for mentioning these points. According to your suggestions, we have corrected these grammar problems.

[Referee]

8. I am interested in the comparison between the NV-based technique and other EO-based electric field measurements, if any. Could the author add some information about this? Typically the NV-based magnetometer could be comparable to the state-of-the-art techniques (see Barry et al, PRApplied (2024) and Wang et al, Nature Communications (2024)). For

single-NV with high resolution also could reach a fairly high sensitivity. Therefore I want to know if the NV-based electrometer could also be comparable with other techniques. This comparison will be of great interest to a broader community.

#Our reply#

Thank you very much for pointing out this issue. We have added a general sensitivity based on measurement time (typically 1 second at 10 Hz scanning frequency) and the noise level of $\Delta R_{eo}/R_0 \approx 2 \times 10^{-7}$ corresponding to $|\Delta E| \approx 9.6 \times 10^{-3}$ V/ μm or 96 V/cm for the EO signal in *n*-GaAs and discuss the resolutions. Using the above parameters we could estimate the sensitivity to be $\sim 1 \times 10^{-2}$ V/ $\mu\text{m}/\sqrt{\text{Hz}}$ or ~ 100 V/cm/ $\sqrt{\text{Hz}}$, which is comparable to Dolde *et al*, Nature Phys. Vol.7, 459 (2011), but three-orders worse than recent work by Michl *et al*, Nano Lett. Vol.19, 4904 (2019). In the revision, we have added a comparison of our method with NV-based technique and other EO-based electric field measurements appropriately.

Response to Reviewer #4

We would like to express our gratitude to the reviewers for reading our manuscript and providing fruitful comments. We have conducted additional experiments after receiving the initial decision of the editors and have revised our manuscript following the reviewers' suggestions.

[Referee]

In the manuscript titled 'Ultrafast diamond nonlinear photonic sensor' by Daisuke Sato et al., Authors present a concept of a nanoscale sensor of electric fields, based on reflective electro-optic response. Here, nitrogen-vacancy (NV) color centers are created in a miniature diamond to break the centrosymmetry and induce non-zero electro-optic (Pockels) effect. With such a diamond probe mounted on a scanning probe microscope, optical readout of local electric fields is proposed, with temporal resolution of 10-fs set by a pump laser pulse. Moreover, Authors propose the use of the same NV diamond for complementary magnetic and temperature measurements, albeit only in conclusions.

Overall, I have found this work novel and of high significance for studies of 2D materials. The high temporal resolution (sub-1 ps) is evident from provided figures. Similarly, the spatial resolution is around or just under 1 micron. I have only some minor remarks regarding the manuscript which I list below.

(1) When reading the manuscript, I was missing a clear explanation that the method relies on reflective EO response. 2. When introducing diamond sensor preparation, a nearly 13-times enhancement is reported, but lacks explanation with respect to what? And how is the optimum NV density defined. Some explanation here is needed.

#Our reply#

Thank you for pointing out the issue. In the previous study, Ichikawa *et al.*, Nat. Commun. **15**, 7174 (2024); Ref. [18], we have measured the dose dependence of the EO response and found that the nearly 13-fold enhancement of the EO response was observed at the N^+ ion dose of $1 \times 10^{12} N^+ cm^{-2}$ with respect to that before ion irradiation (pure diamond).

Furthermore, as presented in Fig. R6 below, the ODMR measurements show the contrast of the NV⁻ resonant dip was maximized at $1 \times 10^{12} N^+ cm^{-2}$, indicating the density of NV⁻ was enhanced. Thus, both ultrafast EO and ODMR measurements support the premise that NV⁻ plays the main role in the enhancement of the EO response. In the revision, we have added these observations to the discussion accordingly.

Fig. R6. Optically detected magnetic resonance (ODMR) spectra obtained at room temperature for the NV diamond samples using 532 nm cw laser. The ODMR measurements show the contrast of the NV⁻ resonant dip was maximized (14.5 %) at the dose of $1 \times 10^{12} N^+ cm^{-2}$, indicating the density of NV⁻ was enhanced at $1 \times 10^{12} N^+ cm^{-2}$.

[Referee]

(2) Equation (1) is introduced without explanation of what is ΔR_{eo} or R_0 . Also, it relates only to material properties of GaAs. Whereas that manuscript section is introduced with mentioning 'evaluation of the nonlinear optical response of the diamond NV probe'. This is confusing for the reader, as the NV-probe EO response is only inferred from Fig. 3c.

#Our reply#

Thank you for this important question. Regarding to Equation (1), $\Delta R_{eo}(t)$ is the anisotropic reflectivity change and R_0 is the reflectivity without photoexcitation. It can be measured by subtracting the two orthogonal component of the photocurrent results in measurement of the anisotropic reflectivity change (please refer Fig. R2 below), a quantity that is proportional to the electric field $\Delta E(t)$,

$$\frac{\Delta R_{eo}(t)}{R_0} = \frac{4}{n_0^2 - 1} (\Delta n_x(t) - \Delta n_y(t)) = \frac{4n_0^3}{(n_0^2 - 1)} r^{ij} \Delta E(t) = \frac{-8}{(n_0^2 - 1)n_0} \chi_{ij}^{(2)} \Delta E(t) \quad (R3)$$

where $n_x(t) = n_0 + \frac{1}{2}n_0^3 r^{ij} E$ and $n_y(t) = n_0 - \frac{1}{2}n_0^3 r^{ij} E$ are the x- and y-components of the indices of refraction, and R_0 is the reflectivity without photoexcitation, n_0 is the static refractive index and r^{ij} is the Pockels coefficient, both of which depend on the material. Eq. (R3) clearly indicates the EO sampling is the basis for the measurement of the change of the electric field $\Delta E(t)$ and this is only possible when $\chi_{ij}^{(2)} \neq 0$, corresponding to the case of NV diamond.

Fig. R2. (a) Electro-optic detection of the anisotropic change in the refractive index. The polarization of the probe was 45° with respect to the optical plane. (b) Isotropic refractive index (n_0) before the photoexcitation ($E=0$) and the anisotropic refractive index (n_{eo}) after the photoexcitation ($E \neq 0$).

Regarding to Fig. 3, Fig. 3b was measured without a NV probe, therefore, the Pockels coefficient should be for n -GaAs ($r_{41} \approx -1.6 \text{ pm/V}$), while in the case of Fig. 3c, we measure the EO response from n -GaAs through the NV probe, that means the Pockels coefficient should be used for NV diamond, which is in general unknown. We recently, however, obtained the value of $\chi_{ij}^{(2)}$ for NV diamond using a second harmonic generation approach as described in Ref. [17], that was $\chi_{ij}^{(2)} \approx 100 \text{ pm/V}$. Using the general relationship of $r^{ij} = -\frac{2}{n_0^4} \chi_{ij}^{(2)}$ (please refer e.g., R. W. Boyd, *Nonlinear Optics*, Academic Press, 2008), we obtain $r^{ij} \approx -6 \text{ pm/V}$ for $n_0=2.4$ (diamond). Note that in general $|r^{33}| > |r^{22}|$ for $3m(C_{3v})$ crystal symmetry (NV diamond) and r^{33} may play a central role, but more theoretical and experimental work is required to fully understand the Pockels coefficient of NV diamond. In the revision, we have added the explanation for equation (1) and the above corresponding discussion appropriately.

[Referee]

(3) Further, Authors claim that 'The signal reduction of $\approx 1/42$ suggests that the probing depth of the diamond NV tip is ≤ 2.4 nm'. How is this depth inferred? How can it be justified given that mean NV centers depth in diamond is on the order of 40 nm (Ref. 18 and Ref. 45 therein) for nitrogen implanted at 30 keV. It's hard for me to imagine NV centers being sensitive only to a 2.4-nm-thin layer while being a 40 nm stand-off distance. I would like to see a comment on this.

#Our reply#

Thank you for pointing out the important consideration. We agree with the statement that it is hard to imagine NV centers being sensitive only to a 2.4-nm-thin layer while being at a 40 nm stand-off distance. We now reconsidered and noticed that it would be more appropriate to focus on the lateral direction. We can estimate the spot size of the focused probe beam through a reflective objective ($f = 13.3$ mm) to be $9.3 \mu\text{m}$, from which the probing area of $21 \mu\text{m}^2$ is obtained. If we think the probing area is limited by the NV probe and this causes a decrease of the EO signal by $1/42$ in Fig. 3, we can estimate the diameter of the NV probe to be $\approx 0.8 \mu\text{m}$, which is nearly consistent with the observation in Fig. 2 and Fig. R5 below. We have revised the corresponding sections based on these new considerations.

Fig. R5. The photoluminescence (PL) image (Top) and the line profile (Bottom) at the horizontal dashed line. Based on the Gaussian fit of the profile, the spatial resolution of the diamond NV probe is better than ≈ 660 nm and potentially ≤ 500 nm because of EO sensitivity enhancement at the apex of the NV tip.

Second, in reply to the reviewer comment, we would like to mention that the probing depth in the sample, i.e., n -GaAs, is larger than 2.4 nm and possibly comparable to a 40 nm stand-off distance as schematically presented in the revised Fig. 1, i.e., Fig. R12 below. In the revision, we have revised these points accordingly.

Fig. R12. (a) The schematic of the “pin-point mode”, that is vertically approaching and retracting the AFM probe at each designated points on the sample. (b) The schematic of electro-optic sampling using a diamond NV tip. The Pockels effect occurs where the spatial inversion symmetry is broken by the NV centers ($\chi^{(2)} \neq 0$), and the refractive index change ($\Delta n(E) \propto \chi_{ij}^{(2)} \Delta E(t)$) of this part is probed. The electric field, which is detected by the NV probe, is generated by the photoexcitation of electrons at the sample surface.

[Referee]

(4) In section Methods, I am missing information on pulses polarization. Is it the same arrangement as in the previous Nat. Commun. paper (Ref. 18)?

#Our reply#

The polarization of the pump and probe beams are similar in the case of the Nat. Commun. paper (Ref. 18), i.e., probe light propagates in the [100] direction and the pump polarization is the $[1 \bar{1} 0]$ direction on the diamond tip. In the revision, we have added this information in the revised Fig. 2a and the Methods section.

[Referee]

(5) I am wondering if the NV sensor concept can be improved by appropriately shaping the diamond. For instance, there are commercially available (Qnami, Qzabre) AFM chips with NV diamond tips. One can already purchase a chip with a single NV center (can be either spatially oriented) or NV ensembles. Such tips can have the NV just a few nm away from the AFM tip apex, bringing it very close to the sample and increasing the spatial resolution to a few nm as well. Additionally, the diamond tip may be parabolically shaped further improving coupling of the probe beam with the NV. Would such chips be a good candidate for the improvements suggested in Conclusions?

#Our reply#

Thank you for the interesting comments. We have actually contacted both Qnami and Qzabre and asked about the possibility of using their beautiful AFM chips for our self-sensing cantilever. Unfortunately, however, it looks difficult to attach their beautiful AFM chips on the self-sensing cantilever because of the less adhesive surface. The current 1st AFM system is based on a self-sensing cantilever system from SENSOR Tech. Inc. This was our choice when we decided upon the order of the custom-made AFM system from Anfatech Inc. We are planning to build a 2nd AFM system based on a tuning fork system, to which a commercial (either Qnami or Qzabre) diamond NV tips can be attached. In the revision, we added a future prospect for using such a good candidate to improve the resolutions.

Summary of the changes made:

1. Page 1, abstract: We have fixed a grammatical issue. (Response to the 7th comment from the Reviewer #3.)
2. Page 2, second paragraph: We have added a general introduction to quantum technologies based on NV centers in lines 43-46. (Response to the 1st comment from the Reviewer #3.)
3. Page 2, second paragraph: We have corrected “millisecond range” into “nanosecond range”. (Response to the 3rd comment from the Reviewer #1.)
4. Page 3, first paragraph: We have added the statement why doesn't try other EO materials or compare their measurements with diamond tips. (Response to the 1st comment from the Reviewer #2.)
5. Page 3, first paragraph: we have appropriately mention that other color centers, such as BV and OV centers, can also break the spatial symmetry. (Response to the 4th comment from the Reviewer #1.)
6. Page 3, first paragraph and Supplementary files: We have added discussion for the difference between our method and super-resolution microscopy (SIM) technique. (Response to the 2nd comment from the Reviewer #2.)
7. Page 3, second paragraph: We have added notes for the contact mode and the height of the NV probe. (Response to the 2nd comment from the Reviewer #1.)
8. Page 3, second paragraph: We have summarized Lines 66-83 into several sentences and moving the other detailed descriptions to the Methods section. (Response to the 1st comment from the Reviewer #3.)
9. Page 4, first paragraph: We have added the statement that the charge state of NV⁻ would play a central role in the EO response. (Response to the 6th comment from the Reviewer #1.)
10. Page 4, first paragraph: We have added the statement about the spatial resolution with diamond tips. (Response to the 2nd comment from the Reviewer #1.)
11. Page 4, second paragraph and Supplementary files: We have added the information for the EO measurements in the Supplementary files. (Response to the 1st comment from the Reviewer #1, the 2nd comment from the Reviewers #2, and the 3rd and 4th comments from the Reviewer #3.)
12. Page 5, first paragraph: We have added explanation for equation (1). (Response to the 3rd comment from the Reviewer #2, and the 2nd comment from the Reviewer #4.)
13. Page 6, first paragraph: We have revised explanation for the decrease in the EO signal when using the NV tip. (Response to the 3rd comment from the Reviewer #4.)
14. Page 6, second paragraph: We have added a note regarding Raman measurements. (Response to the 4th comment from the Reviewer #2.)
15. Page 9, third paragraph: We have added a discussion regarding the Stark effects. (Response to the 6th comment from the Reviewer #3.)

16. Page 10, second paragraph: We have added a discussion regarding the sensitivity of our measurements. (Response to the 8th comment from the Reviewers #3.)
17. Page 10, second paragraph: We have added a discussion for the possible use of commercial (Qnami or Qzabre) diamond NV tip. (Response to the 2nd comment from Reviewer #2 and 5th comment from the Reviewer #4.)
18. Page 10, second paragraph: We have added a discussion for the possible experiment of vertical distribution of the electric field. (Response to the 5th comment from Reviewer #1.)
19. Page 11, Method section: We have moved the details of the fabrication of nano probe from the main text. Here the explanation for the nearly 13-times enhancement with respect to that before ion irradiation (pure diamond) has been added. (Response to the 1st comment from the Reviewer #3 and the 1st comment from the Reviewer #4.)
20. Page 11, Method section: We have added the details of the polarization with respect to the diamond NV probe. (Response to the 4th comment from the Reviewer #4.)
21. Page 12, Method section: We have added the details of the photodetector used. (Response to the 3rd comment from the Reviewer #2.)
22. Page 13-15, References: We have added important citations. (Response to the 5th comment from the Reviewer #3.)
23. Fig. 1 has been revised. (Response to the 2nd comment from the Reviewer #1, the 2nd comment from the Reviewer #3, and the 3rd comment from the Reviewer #4.)
24. Fig. 2 was revised to make the part (e) clearer. (Response to the 2nd comment from the Reviewer #2.)
25. Fig. 4, caption: We have added the notes for the fluctuation of height in Figure 4c. (Response to the 4th comment from the Reviewer #2.)
26. We have revised the format based on the guidelines including references.

Response to Reviewer #2:

We are grateful to the reviewer for reviewing our manuscript again. Although the reviewer was not satisfied with the initial revision, we feel that the reviewer's comments are addressed to additional or future work and do not address the current results. In our reply, we have addressed possible future work in a few sentences in an additional discussion section in the revised version.

[Referee]

In this article, the authors proposed a high spatial and high time resolution electric field measurement method based on femtosecond lasers and diamond scanning probe. After the author modified the paper substantially, I still do not recommend their work for publication in the Nature Communications journal. There are still some comments about this work as follows:

(1). The author admits that NV center is used to break the spatial inversion symmetry of diamond to probe the electro-optic (EO) effect instead of magnetic and even temperature sensing in this work. And there are indeed lots of semiconductor transparent materials with a significant non-zero second-order nonlinear susceptibility, which can detect the electro-optic (EO) effect very accurately. Furthermore, there is a paramagnetic defect in GaN which has similar quantum sensing performance (See Luo, Jialun, et al. "Room temperature optically detected magnetic resonance of single spins in GaN." Nat. Mater. 23, 512-518 (2024)). The GaN materials should be used for probing the electro-optic (EO) effect, magnetic field and temperature instead of diamond materials.

#Our reply#

We must stress again that we have indeed demonstrated a high spatial and time resolution electric field measurement method based on a femtosecond laser and a NV color center scanning probe, as evaluated by the other reviewers. Diamond materials are still an important platform for the community of quantum sensing. EO experiments using a new system (GaN etc.) are outside the scope of the current manuscript and should be addressed in a future work. Thus, we disagree with the reviewer's statement that "The GaN materials should be used for probing the electro-optic (EO) effect, magnetic field and temperature instead of diamond materials".

[Referee]

(2). The experimental content of investigation WSe₂ films with diamond is too few to form a systematic and clear physical viewpoint. Transition metal dichalcogenides are a promising family of materials for electronics and optoelectronics, in part due to their range of bandgaps that can be modulated by layer number. And WSe₂ can be selectively grown with one, two, or three layers, as regulated by a one-step hydrogen-controlled chemical vapor deposition (H-CVD) process involving cyclical pulses of H₂ flow (See DeGregorio, Zachary P., Jason C. Myers, and Stephen A. Campbell. "Rational control of WSe₂ layer number via hydrogen-controlled chemical vapor deposition." Nanotechnology 31, 315604(2020)). The electro-optic (EO) effect of the resulting mono-, bi-, and tri-layer WSe₂ films should be investigated in this work. Furthermore, I think the substrate have an impact on the physical properties of WSe₂ films and different substrates should be investigated in experiment instead of Si substrate only.

#Our reply#

We must stress that EO measurements of mono-, bi-, and tri-layer of WSe₂ films on different substrates is beyond the scope of the current manuscript and is a possible topic of a future work. In the revision, we have added possible future samples.

[Referee]

(3). In the supplementary note 4, the density of NV- centers in diamond can be enhanced by high energy electron beam irradiation (See Capelli, Marco, et al. "Increased nitrogen-vacancy centre creation yield in diamond through electron beam irradiation at high temperature." Carbon 143, 714-719(2019)). Hence, I think the intensity of EO signal will be improved with this method and should be investigated in the experiment.

#Our reply#

The improvement of the EO signal by electron beam irradiation is well beyond the scope of the current manuscript; we may consider such an investigation in a future work if we can find a collaborator who has a high energy electron beam irradiation system.

[Referee]

(4). Since the quantum coherence of NV center is not employed in this work (Degen, Christian L., Friedemann Reinhard, and Paola Cappellaro. "Quantum sensing." Rev. Mod. Phys. 89, 035002(2017).), the terminology of quantum sensing or quantum device based on NV center in the diamond should be deleted or modified to get rid of misleading during academic exchanges.

#Our reply#

The quantum coherence of NV center as demonstrated in previous papers is outside the scope of the current manuscript. Inclusion of “quantum (NV) sensing” in the introduction was potentially suggested by another reviewer. There is no use of the terminology of quantum sensing or quantum device in the Results section. So, we will not delete these words from the introduction.

Response to Reviewer #3:

We appreciate the review of our manuscript again and the fruitful comments. We have further revised our manuscript following the reviewer's suggestions.

[Referee]

The authors have made dramatic revisions to both my comments and those of other reviewers. I am satisfied with all the replies for the scientific arguments. As I mentioned in the previous review, with proper revisions, this work will have a significant impact on a broad audience. However, after re-reading the manuscript carefully, I still have several minor suggestions. I am hesitant about whether another round of review will be needed before formally recommending the publication, but at the very least, I strongly encourage the authors to further revise the manuscript to improve clarity and maximize its potential to attract interest and citations.

1. There are still several typos. For example, in line 83, it should read "Diamond nonlinear photonic sensor" rather than "photonics sensor." Additionally, the acronym SIM is defined inconsistently: it stands for Structured Illumination Microscopy on line 65, but refers to Scanning Ion Microscopy on line 92. The authors should revise the text to resolve this inconsistency.

#Our reply#

Thank you for kindly pointing out the typos. We have corrected the typo on line 83 accordingly. Regarding the comment on line 92, Scanning Ion Microscopy is the correct phrase. We have deleted "SIM" for Scanning Ion Microscopy to avoid confusion. In addition, we checked again over the manuscript to check for other typographical errors.

[Referee]

2. In line 46, I do not believe that "quantum emitters" constitute an "application" on the same level as quantum computing and quantum communication. I suggest using the term "quantum networks" instead, as it more accurately represents one of the current state-of-the-art applications of quantum emitters. Personally, I would not recommend placing emphasis on the potential application of NV centers in quantum computing, as there are still many unresolved challenges. Moreover, there are already numerous misconceptions and misleading terms in this area, which have caused considerable confusion among audiences. That said, this is merely my personal opinion, and the final decision ultimately rests with the authors.

#Our reply#

Thank you for the suggestions. After careful reading of relevant papers on quantum computing, we recognize the reviewer is right at the point that there are still many unresolved challenges regarding this issue. In the revision, we have replaced "quantum emitters" by "quantum networks" and "quantum computing" by "quantum information processing".

[Referee]

3. Following my last review, I have suggestions for the authors regarding the paper citations, as several citations still do not align with the contents of the author's manuscript. For example, Ref. 16 describes the high-fidelity transfer and storage of information using nuclear spins, which does not necessarily pertain directly to quantum communication. Several other works may be more relevant in this context. For instance, Pompili et al., Science (2021), and Knaut et al., Nature (2024), report on the development of a multimode quantum internet using remote NV and SiV centers. Additionally, Wang et al., Nature Communications 14, 704 (2023), present advancements in quantum repeater and quantum router architectures for

remote quantum communication. I suggest that the authors consider citing these more pertinent works.

The authors mentioned group-IV vacancies but also do not add appropriate citations. Ref. 26 does not show the inversion symmetry etc. A much more appropriate citation will be Hepp et al, PRL (2014) for SiV.

I suggest the authors to check all the referred papers again.

#Our reply#

Thank you for so kind suggestion for improving the references. We have replaced Ref. 16 and 26 by more pertinent works as the reviewer suggested. In addition, we have double-checked that all the referred papers are appropriate.

[Referee]

4. What are BV, OV centers on line 64? There is no definition on this.

#Our reply#

Thank you for this important question. BV, OV centers are boron-vacancy and oxygen-vacancy defects, respectively. Accordingly, in the revision, we have added the definitions.

[Referee]

5. I think line 73 may be more suitable for Methods section, like AFM pin point mode may be too detailed in the main text, especially in the introduction session.

#Our reply#

Thank you for the fruitful suggestion. We have moved line 73 into the Methods section as the reviewer suggested.

[Referee]

6. The introduction session is not well-written. The connection between the second and third paragraph is vague. What are the points to mention other quantum defects in line 61-63 and why mentioning the reason not to use other EO crystals? Those discussions are more suitable to the conclusion but not introduction. The readers will not care what are not describing in the manuscript in the introductions. The current version is misleading and could be improved in a further revision.

#Our reply#

Thank you for suggesting possible improvements in the writing of the Introduction. We have rewritten the latter part of the introduction by moving lines 57-64 to the Methods section.

[Referee]

7. To reiterate, I have never questioned the scientific value of the manuscript. However, I strongly recommend that the authors carefully revise the paper once more and correct the remaining typographical errors before publication. Since it will potentially be published in a prestigious journal like Nature Communications, it is important to maintain a high standard of clarity, conciseness, and overall writing quality.

#Our reply#

Thank you again for carefully reading the manuscript and offering fruitful comments. After addressing all of comments, we hope the revised version is easier to follow.

Response to Reviewer #4:

We appreciate the review of our manuscript again and the fruitful comments. We have further revised our manuscript following the reviewer's suggestions.

[Referee]

In the amended manuscript, Authors have addressed points raised in my review, as well as over Reviewers' ones. In particular, they explained now much better the principles of measuring the EO response. I am content with the changes made and sure these make the manuscript easier to read.

1. The only left over remark I have is the missing discussion of the implantation depth and thickness of the NV-layer created and used for the experiment.

For non-expert readers, it may be difficult to assess how the NV-layer looks like if it's characterized only by the concentration or ion fluence. Moreover, the sentence: "Thus, the height of the NV center probe was quasi-zero at each of the designated 76 points, which was determined by the Lennard-Jones potential (Typically 0.1-0.3 nm)" is misleading since it discusses the distance of the diamond (plate) rather than NVs (layer in that plate) from the surface.

#Our reply#

Thank you for kindly pointing out these problems. First, the implanted nitrogen has an average depth of ~40 nm and is distributed like a Gaussian with a full width at half maximum (FWHM) of ~25 nm, as demonstrated in Fig. R1 which shows the results of a SRIM simulation [Ziegler, J. F., Ziegler, M. D. & Biersack, J. P. SRIM - the stopping and range of ions in matter. Nucl. Instrum. Methods Phys. Res. Sect. B **268**, 1818–1823 (2010)].

Second, to avoid the inclusion of a possibly misleading sentence, we have corrected the sentence: "Thus, the distance of the diamond probe from the sample surface was quasi-zero at each of the designated points, which was determined by the Lennard-Jones potential (Typically 0.1-0.3 nm)".

Fig. R1. Distribution of nitrogen ions (blue dots) and vacancies (red dots) as a function of implantation depth, simulated by SRIM for the surface of (100) diamond for an implantation energy of 30 keV and an incident angle of 7°, with the fluence of 1×10^{12} ions/cm². The dashed line represents a fit using a Gaussian function.

[Referee]

2. Apart from this issue, I believe the article is appropriate for publication in Nature Communications, as it brings attention to a novel and interesting application of NV diamonds.

#Our reply#

Thank you again for carefully reading the manuscript and offering fruitful comments. After addressing all of comments, we hope the revised version is easier to follow.

Summary of the changes made:

1. Page 2, second paragraph: We have corrected the words “quantum emitters” and “quantum computing”. (Response to the 2nd comment from the Reviewer #3.)
2. Page 3, first paragraph: We have moved line 73 into Methods section. (Response to the 5th comment from the Reviewer #3.)
3. Page 3, second paragraph: We have reconstructed the last part of the introduction by moving line 57-64 to the Methods section. (Response to the 6th comment from the Reviewer #3.)
4. Page 3, third paragraph: We have corrected a typo on the sub-title. (Response to the 1st comment from the Reviewer #3.)
5. Page 4, first paragraph: We have deleted “SIM” for scanning ion microscopy to avoid the confusion. (Response to the 1st comment from the Reviewer #3.)
6. Page 9, first paragraph: We have added possible future samples. (Response to the 2nd comment from the Reviewer #2.)
7. Page 10, Methods section (Fabrication of nano probe): We have added the information for the implanted depth and profile of $^{14}\text{N}^+$ ions. (Response to the 1st comment from the Reviewer #4.)
8. Page 11, Methods section (Fabrication of nano probe): We have added the definitions for BV, OV centers. (Response to the 4th comment from the Reviewer #3.)
9. Page 12, Methods section (Ultrafast nanoscopy): We have corrected the sentence regarding to the distance of the diamond probe from the surface. (Response to the 1st comment from the Reviewer #4.)
10. Page 13-15, References: We have revised several papers to make the citation more appropriate. (Response to the 3rd comment from the Reviewer #3.)